



# High-frequency Internal Waves, High-mode Nonlinear Waves and K-H Billows on the South China Sea's Shelf Revealed by Marine Seismic Observation

Linghan Meng[1,2], Haibin Song[1,2], Yongxian Guan[3], Shun Yang[1,2], Kun Zhang[1,2], Mengli Liu[1,2]

[1] School of Ocean and Earth Science, Tongji University, Shanghai, 200092, China

[2] State Key Laboratory of Marine Geology, Tongji University, Shanghai, 200092, China

[3] MNR Key Laboratory of Marine Mineral Resources, Guangzhou Marine Geological Survey, China Geological Survey, Guangzhou, 510760, China

**Correspondence:** Haibin Song (hbsong@tongji.edu.cn) and Yongxian Guan (813385148@qq.com)

**Abstract.** From July to September 2009, a set of multi-channel seismic data was collected in the northern shelf area of the South China Sea. After the data was processed, we observed a series of shoaling events on one of the survey lines, including high-frequency internal waves, high-mode nonlinear internal waves, and shear instability. Using theoretical results from previous numerical simulations and field observations, coupled with local temperature and salinity data, we analyzed their depth distribution, waveform characteristics, and formation mechanisms, and discussed the influence of seafloor topography and stratification on the shoaling of solitary internal waves. We estimated the mixing parameters of seawater using a parameterization scheme based on hydrographic data and seismic data, respectively. And we found that the diapycnal mixing caused by these shoaling events in the shelf area were about 3.5 times greater than those on the slope. Consequently, the fission of internal solitary waves and the induced shear instability serve as significant mechanisms for the energy dissipation of internal solitary waves at the slope and shelf of the South China Sea. Additionally, the high-frequency internal waves generated during shoaling might also have a crucial role in this process.

## 1 Introduction

Internal solitary waves are widely distributed in the global ocean, and the South China Sea is recognized as one of the best places to study large-amplitude internal solitary waves (Klymak et al., 2006; Zheng et al., 2022; Cai et al., 2015). After years of research, there is a clearer understanding of their generation, propagation, and evolution: barotropic tides interact with complex topography to produce internal tides (baroclinic tides), strong internal tides radiate out from the Luzon Strait, with those radiating westward crossing the northern South China Sea basin and propagating towards the slope and shelf areas. During propagation, nearly half of the internal tide energy generates internal solitary waves through nonlinear steepening, and these waves deform due to shoaling effects, eventually breaking and dissipating on the shelf (Alford et al., 2015; Bourgault et





al., 2007; Fu et al., 2012; Sinnett et al., 2022; Liu et al., 2022). These large-amplitude internal solitary waves have significant
impacts on ocean mixing, sediment suspension, nutrient transport, and offshore oil and gas engineering (Bogucki et al., 1997;
Osborne et al., 1978; Wang et al., 2007; Xu and Yin, 2011).

Internal solitary waves undergo shoaling processes when they propagate from deep water to the slope and shelf areas.
Moorings and high-frequency acoustic instruments can record this evolution process in detail (Bourgault et al., 2007; Fu et al.,
2012; Orr and Mignerey, 2003; Xu and Yin, 2012). Typically, as they reach critical depths, these waves steepen at the rear,
reversing polarity to form elevated waves. On gentle slopes, dispersion continues to form elevated wave trains, a process called
fission (Bai et al., 2019; Djordjevic and Redekopp, 1978; Gong et al., 2021b; Zheng et al., 2001). Additionally, Bai et al. (2019)
found that internal solitary waves also undergo fission to produce high-frequency internal waves during shoaling in the South
China Sea shelf area. On steeper slopes, waves break directly (Aghsaee et al., 2010). These processes lead to shear or
convective instabilities, causing significant turbulence and mixing. The former is often in the form of K-H waves at the trough
and its trailing edge or within the seawater layers (near the thermocline) (Chang et al., 2016; Lamb and Farmer, 2011; Moum
et al., 2003; Terletska et al., 2018; Van Haren and Gostiaux, 2010), while the latter forms an internal solitary wave with a
"trapped vortex core" (Lien et al., 2012). However, traditional marine observational instruments often fail to clearly capture
the features of internal waves during shoaling, including their waveforms, propagation speeds, and evolution process, as well
as the resulting energy dissipation and enhanced mixing in the ocean, due to spatiotemporal limitations. The shoaling process
of internal waves is usually studied through numerical simulations, which can significantly differ from actual seawater
conditions.

Seismic oceanography, initially proposed by Holbrook (2003), primarily uses reflection seismic data to study physical
oceanographic phenomena. Due to its advantages of fast acquisition, high horizontal resolution, and full-depth water column
imaging, this method has been extensively used in the study of internal solitary waves (e.g., Tang et al., 2014), eddies (e.g.,
Yang et al., 2022), and fronts (e.g., Gunn et al., 2021). In this paper, we attempt to use this method to study the shoaling of
internal solitary waves and were fortunate to observe the high-frequency internal waves, high-mode nonlinear internal waves,
and induced shear instability (Sect 3.1). We then investigate their formation mechanisms, waveform characteristics, etc. (Sect
3.2, Sect3.3, Sect 3.4), and estimate the induced turbulence dissipation rate and diapycnal mixing (Sect 3.5, Sect 3.6), Finally,
we also discuss the dissipation mechanism of internal solitary waves in the South China Sea (Sect 4).

## 2 Data and methods

### 2.1 Seismic data acquisition and processing

From July to September 2009, the Guangzhou Marine Geological Survey Bureau conducted multi-channel reflection seismic
data collection on the northern slope and shelf of the South China Sea, covering 43 lines (not shown). Our study focused on
Line 25 (red solid line in Fig. 1a). Acoustic signals were generated by a 5080 in³ (83 L) air gun array, fired every 25 m. The
main frequency of the seismic signals was 35 Hz. A 6 km streamer with 480 hydrophones, spaced 12.5 m apart and with a



minimum offset of 250 m, was towed to receive reflections, sampling every 2 ms.

To clearly image the water layer, standard processes were applied to marine seismic data, including 1) defining the observation system; 2) denoising and direct wave suppression; 3) velocity analysis; 4) NMO correction; 5) stacking; 6) further denoising; and 7) migration. A key step is using median filtering and match subtracting in shot gathers to suppress the direct wave, avoiding their strong energy overshadowing the reflections and enhancing the imaging quality of the water layer.

Secondly, to ensure the extracted reflectors accurately represent isopycnal displacement, further denoising (step 6) is required: band-pass filtering is applied to ambient noise, and notch filtering to harmonic noise, ensuring the seismic frequency band carries the best possible turbulence information. Holbrook et al. (2013) used the signal-to-noise ratio of adjacent traces to assess the optimal filtering range, calculated as

$$\frac{S}{N} = \sqrt{\frac{|c|}{|a-c|}} \ , \tag{1}$$

where $c$ is the maximum cross-correlation coefficient of adjacent traces, and $a$ is the autocorrelation coefficient of the first trace. The median S/N value across all traces determines the final result for the profile. Upon verification, we used an 8-12-75-85 Hz band-pass filter to suppress ambient noise. Additionally, harmonic noise from shots appears periodically in the horizontal wavenumber domain $k=k_s$ ($k_s=n/\eta$, where $n$ is an integer, $\eta$ is the shot spacing) as pulses. With a 25 m shot spacing, we effectively suppressed the harmonic noise near $k = 0.04$ m$^{-1}$, 0.08 m$^{-1}$, etc., using appropriate notch filters. This denoising process improved the S/N ratio to 9, well above the minimum standard of 4 (Holbrook et al., 2013).

Finally, assuming a seawater speed of 1500 m/s (with variations from 1480-1540 m/s), the Stolt migration method quickly reveals the true form and position of reflectors, facilitating subsequent reflector picking. The final result of seismic Line 25, as shown in Fig. 2c, exhibits more continuous reflectors compared to former processing (Fig. 2a). Reflections in the seismic section are caused by variations in acoustic impedance, primarily controlled by changes in sound speed (90%-95%), which depend on the seawater temperature gradient (80%). Therefore, the seismic profile essentially represents a high-resolution snapshot of the ocean's vertical temperature gradient (Ruddick et al., 2009; Sallarès et al., 2009). According to Fig. 2c, above 500 m depth (especially between 100-400 m), clear reflector layers indicate strong thermohaline gradients. Below 500 m, as depth increases, the reflection signals gradually weaken or even disappear, suggesting more uniform seawater.



**Figure 1.** Topography of the South China Sea and seismic survey lines. (a) Topographic map of the research area, the black solid line represents the line 25, red dots are XBT data (numbered from right to left, XBT1-XBT9), magenta dots are CTD data (numbered from right to left as CTD1-CTD4); (b) Seabed topography along line 25, calculated to have a seabed slope $\gamma=1.08°$ (0.018).



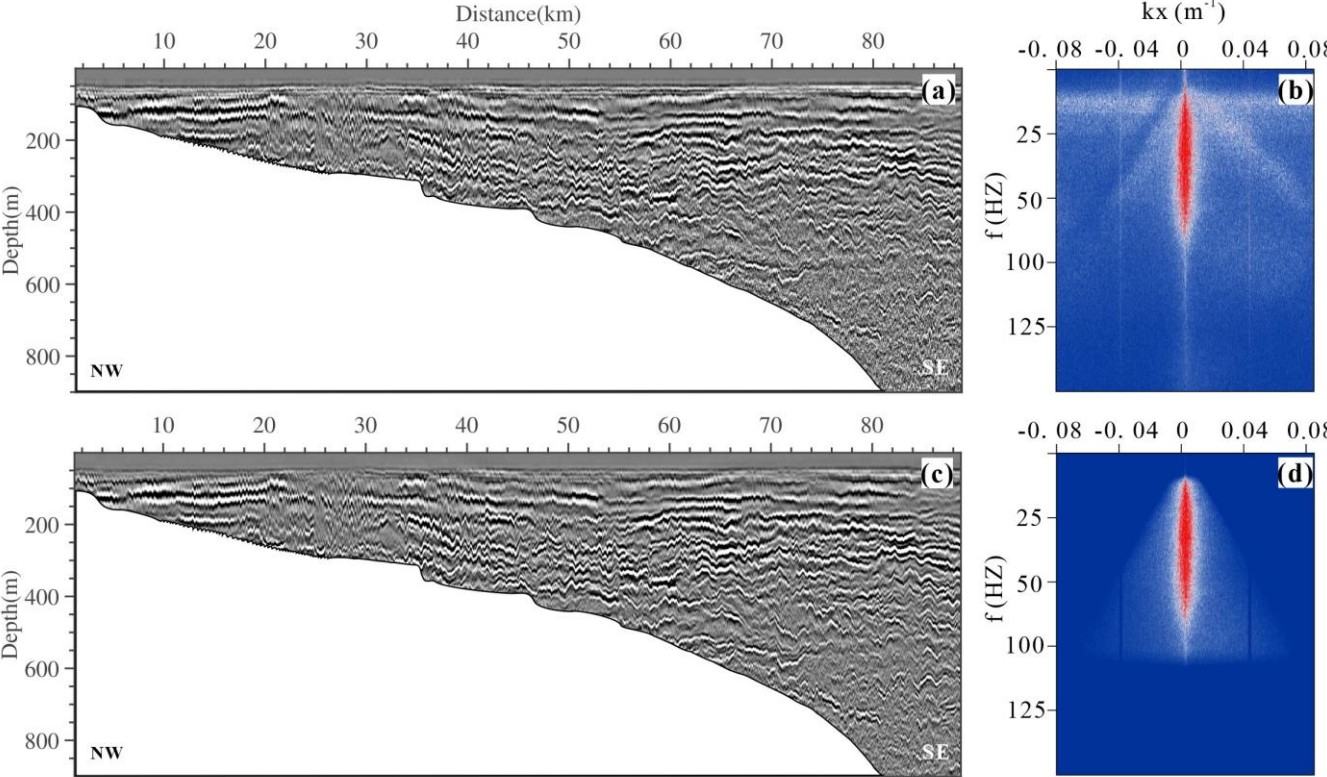

**Figure 2.** Seismic profiles before and after processing. (a) and (b) are the seismic profile and F-K spectrum before processing, respectively; (c) and (d) are the seismic profile and F-K spectrum after processing, respectively.

## 2.2 The breaking of shoaling internal waves

Aghsaee et al. (2010) classified the breaking of internal solitary waves during shoaling into four types: plunging breakers, surging breakers, collapsing breakers, and fissions. Because the breaking of internal waves is directly related to the topography and wave shape, the internal Iribarren number ($\xi_{in} = S/\sqrt{a/L_w}$) is often used to characterize the type of internal wave breaking, where $S$ is the slope of the terrain, $a$ is the initial wave amplitude, and $L_w$ is the initial wave half-wavelength. Among the four types of breaking, the first three tend to occur on steeper seabed slopes, corresponding to high Iribarren numbers, while fission is the most common type of breaking in the ocean, corresponding to low Iribarren numbers, because the seabed slope in shelf areas and near the coast is generally gentle (S≤0.05). In the gentle shelf areas, due to dispersion, internal solitary waves undergo fission, gradually evolving into a series of elevation waves (packets) at the rear of the initial wave, which have enough time to contribute to turbulent mixing during propagation (Masunaga et al., 2019).

Internal wave breaking is the result of hydrodynamic instability in seawater. The breaking often starts at the trailing edge of the wave, that is, where the velocity of the water particles exceeds the phase velocity of the wave (Kao et al., 1985). Vlasenko and Hutter (2002) provided a breaking criterion for strong nonlinear internal waves based on numerical simulations. This




criterion relates the water depth at the location of wave breaking to the initial wave amplitude and the seabed slope:

$$\bar{a} = \frac{a_m}{H_b - H_m} = \frac{0.8^\circ}{\gamma} + 0.4, \tag{2}$$

where $a$ is the dimensionless amplitude, $a_m$ is the initial wave amplitude, $H_m$ is the water depth at the unperturbed pycnocline (the depth of the upper seawater of the initial wave), $\gamma$ is the seabed slope, and $H_b$ is the water depth at the breaking location of the wave. According to equation (2), if the water depth $H_s$ at the shelf is less than $H_b$, the internal solitary wave will break

on the slope or shelf area; otherwise ($H_s > H_b$), it will undergo dispersion and form a series of elevation waves (i.e., fission).

**2.3 Shear instability within the ocean**

Internal solitary waves can undergo shear instability during the shoaling process and often accompany the generation of Kelvin-Helmholtz (K-H) billows, which can propagate downstream after formation, causing intense turbulent dissipation (Geyer et al., 2010; Geyer et al., 2017; Van Haren and Gostiaux, 2010). As shown in Fig. 3, K-H billows typically have an alternating

braid-core structure, resembling "cat's eyes" and often exhibit small-scale secondary instabilities on the braids (Chang et al., 2016; Thorpe, 1987; Tu et al., 2022). It is rare to see a complete "cat's eye" structure in the ocean; more often, they manifest as the braid structures shown in Fig. 3a. Shear instability can be quantitatively assessed by the Richardson number $Ri = N^2/S^2$ (Lamb, 2014), where N is the buoyancy frequency ($N = \sqrt{-(g/\rho)\partial\rho/\partial z}$, g is the acceleration due to gravity, $\rho$ is the density, and $S$ is the vertical shear of the horizontal flow velocity ($S = \sqrt{(\partial u/\partial z)^2 + (\partial v/\partial z)^2}$), with $u$ and $v$ being the components of the horizontal flow velocity, and $z$ is the vertical coordinate. When this parameter is less than 0.25, the presence

of strong shear or weak stratification is conducive to the occurrence of instability. Thorpe (1973) used $Ri_o$ to represent the minimum Richardson number at which instability occurs, which can link field observations with linear instability theory, numerical simulations, and laboratory experiments. However, $Ri_o$ is difficult to obtain directly and often requires the help of the Aspect ratio. The Aspect ratio is the ratio of wave height to wavelength, equivalent to steepness, and it has been found

through laboratory experiments and numerical simulations that its relationship with $Ri_o$ satisfies the empirical formula (Fritts et al., 2011; Thorpe, 1973; Tu et al., 2022):

$$Ri_o = 0.25 - 0.39 h_{es}/\lambda, \tag{3}$$

where $h_{es}/\lambda$ is the ratio of wave height to wavelength, that is, the Aspect ratio. The height and wavelength of the waves can be directly obtained from seismic profiles. Although the height and wavelength often change, this ratio does not vary much.

Tu et al. (2020) summarized more than 10 cases and found that the Aspect ratio in the ocean ranges from 0.08 to 0.31, with an average of 0.15 and a standard deviation of 0.08.





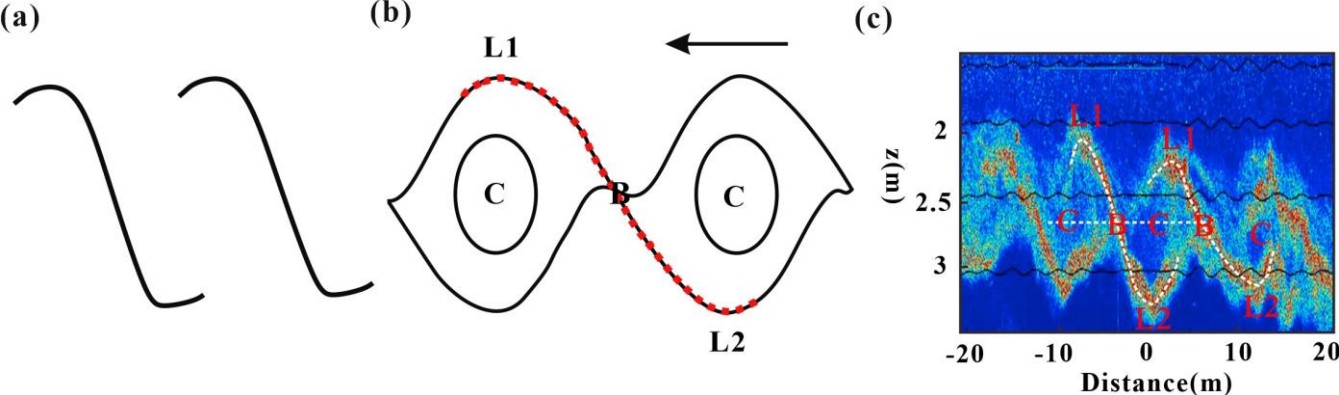

**Figure 3.** Schematic of the K-H billow structure. (a) K-H braids; (b) Cat's eyes, characterized by the alternating upper eyelid of the front billow (L1) - lower eyelid of the following billow (L2) - billow core (C) - braid (B); (c) Geyer et al. (2010) observed K-H billows through an acoustic profile.

## 2.4 Diapycnal diffusivity estimates from seismic data

Klymak and Moum (2007) used horizontal wavenumber spectra from oceanic horizontal towed measurements to estimate the diapycnal diffusivity ($K_\rho$) in seawater. In the open ocean, the horizontal wavenumber spectrum ($\phi_\zeta$) can be clearly divided into two parts: the low wavenumber part is related to internal waves ($\phi_\zeta^{IW}$), with spectral characteristics consistent with the Garrett and Munk (1975) model (GM75), and proportional to the -2.5 power of the wavenumber; the high wavenumber part is dominated by turbulence ($\phi_\zeta^T$), exhibiting Kolmogorov-like behavior (proportional to the -5/3 power of the wavenumber).

Here, we are more interested in the turbulent segment of the slope spectrum. The horizontal wavenumber spectrum of the turbulent segment can be represented by a simplified Batchelor (1959) model (equation 4), so the turbulence kinetic energy dissipation rate ($\varepsilon$) can be obtained from the horizontal wavenumber spectrum. According to equation 5, the diffusivity can be obtained (Osborn, 1980):

$$\phi_\zeta^T = \frac{4\pi\Gamma}{N^2} C_T \varepsilon^{2/3} (2\pi k_x)^{-5/3}, \tag{4}$$

$$K_\rho = \Gamma\varepsilon/N^2, \tag{5}$$

where $\Gamma \approx 0.2$ is the mixing efficiency of seawater, $C_T \approx 0.4$ is considered a constant, and $N$ is the average buoyancy frequency at the corresponding water depth.

Assuming that the isopycnal surfaces of seawater coincide with the seismic reflection layers (Holbrook et al., 2013; Sheen et al., 2009), the horizontal wavenumber spectra obtained from the vertical displacement of seismic reflection layers can replace those obtained from horizontal towed measurements, thus the turbulence dissipation rate and diapycnal diffusivity can also be calculated from seismic data (Dickinson et al., 2017; Gong et al., 2021b; Tang et al., 2021; Yang et al., 2023). To obtain the spatial distribution of the mixing parameters, we gridded the seismic profile of line 25, with each window size being 5 km×75 m (length×width), the lateral step size 2.5 km, and the vertical step size 37.5 m. An automatic picking algorithm was



used to identify seismic reflection layers within the window, each reflection layer being no less than 1 km in length, totaling 410 (Fig. 4a). The vertical displacement of the picked reflection layers is obtained by subtracting their linear fit curves, and then the power spectral density of the displacement curve, i.e., the horizontal wavenumber spectrum ($\phi_\zeta^T$), is calculated using a Fourier transform. The spectral calculation process uses a 128-point sampling point width and a non-overlapping Hanning window.

To better distinguish internal wave with turbulent segments, the displacement spectrum ($\phi_\zeta^T$) is usually multiplied by $(2\pi k_x)^2$ to obtain the slope spectrum ($\phi_{\zeta x}^T$), so the slope of the turbulent segment changes from -5/3 to 1/3, and the slope of the internal wave segment changes from -5/2 to -1/2 (Holbrook et al., 2013). To ensure the stability of the results, the least squares method is used to fit the slope spectrum of the turbulent wavenumber segment (0.0075-0.0378 cpm) to obtain the fitted spectrum (red dashed line). Finally, the fitted spectrum is substituted into equations 3 and 4 to obtain the average diapycnal mixing of seawater (Fig. 4b, 4c, 4d). Additionally, to eliminate the influence of seawater stratification ($N$) on the internal wave field, we need to normalize the slope spectrum according to the local average buoyancy frequency, i.e., multiply by $N/N_0$, where $N$ is derived from Fig. 5d, $N_0$=3 cph.

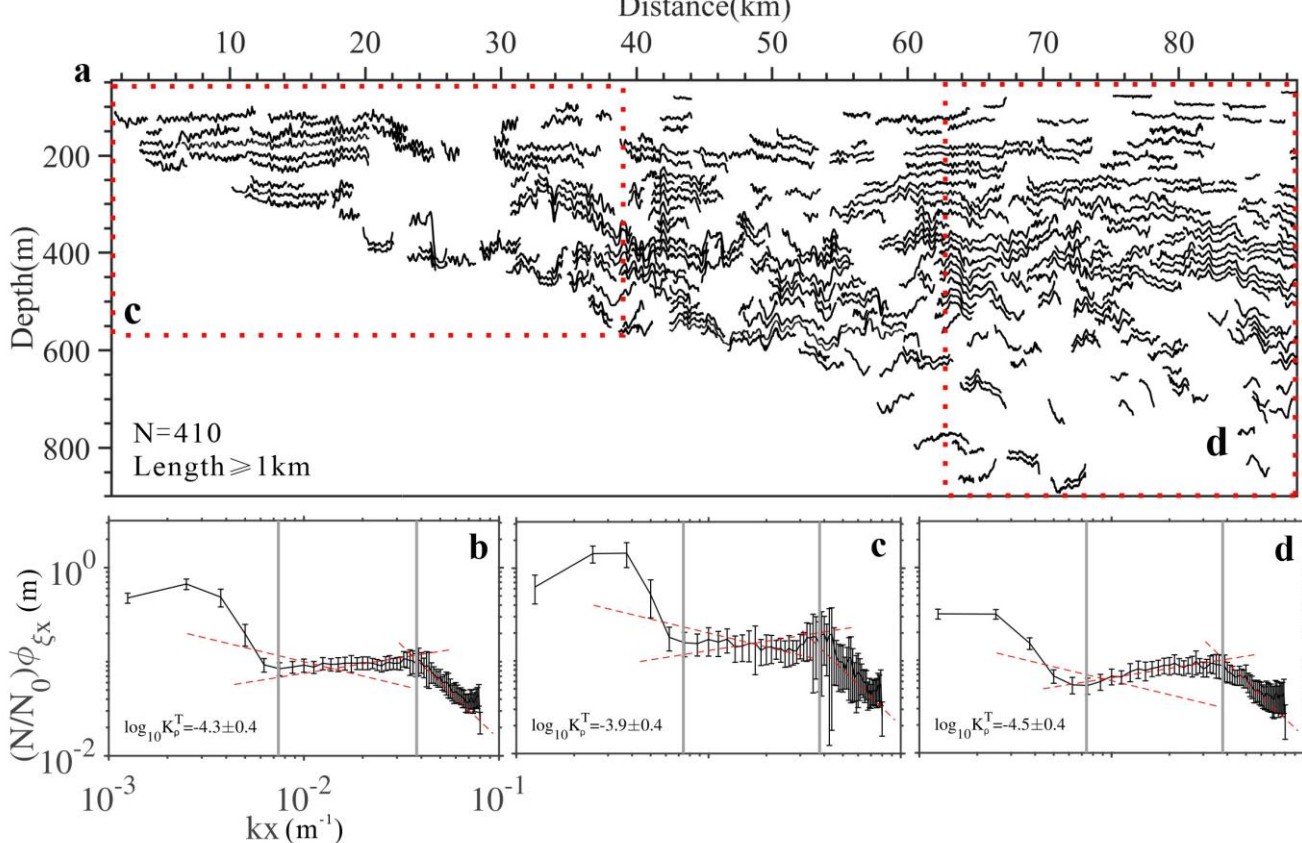

**Figure 4.** Picking of seismic data reflection layers and slope spectrum. (a) Reflection layers picked from the entire seismic profile; (b) Slope



spectrum calculated from all reflection layers in Fig. 3a, with a 95% confidence interval, the red dashed line is the fitted spectrum; (c) and (d) are the slope spectra calculated from the reflection layers in windows c and d of Fig. 3a, respectively, with a 95% confidence interval, the red dashed line is the fitted spectrum. All slope spectra are calibrated with the local buoyancy frequency.

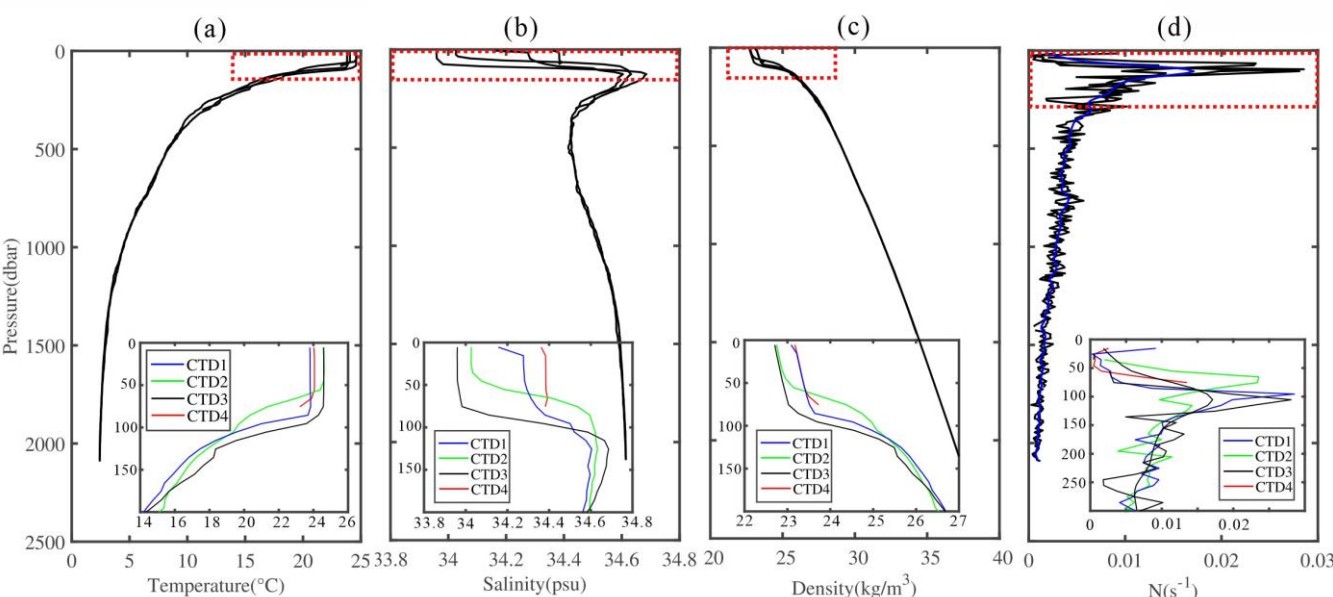

**Figure 5.** Data for temperature (a), salinity (b), density (c), and buoyancy frequency (d) from 4 CTD stations (see station distribution in Fig. 1b). Insets in each graph (a, b, c, d) show data from a depth of 200-300 m within the red dashed-line boxes. The thick blue solid line in graph (d) represents the average buoyancy frequency of seawater.

## 3 Results

### 3.1 Seismic images

Affected by factors such as water depth, topography, and stratification, a variety of changes in waveforms occur during the shoaling process of internal solitary waves, such as the formation of high-frequency internal waves through shoaling, the breaking of internal waves when the slope is steep, the bores formed after breaking containing enclosed cold water, the high-mode internal waves and accompanying K-H billows that appear after interacting with the terrain, etc. (Scotti et al., 2008; Sinnett et al., 2022). Figure 6a is a part of the seismic section (Fig. 2c), with a horizontal range of 1.25-38.75 km and a depth range of 0-300 m. On the left side of the section (horizontal range 1.25-20 km), we can see a series of high-frequency internal waves (Fig. 6b), which are generally small-amplitude, high-frequency internal waves. Adjacent to the high-frequency internal waves on the right side (horizontal range 20-26 km) are two "convex" structure reflectors with larger amplitude and wave width (Fig. 6c). In the horizontal range of 26.5-30 km, K-H billows with a more distinct braid-like structure, shaped like an inverted



"S" can be seen. On the far-left side of the section (horizontal range 33-35 km), a relatively intact and symmetrical mode-2 internal solitary wave can be observed, with relatively small amplitude and wave width. Here, we mainly discuss the first three types of marine phenomena.

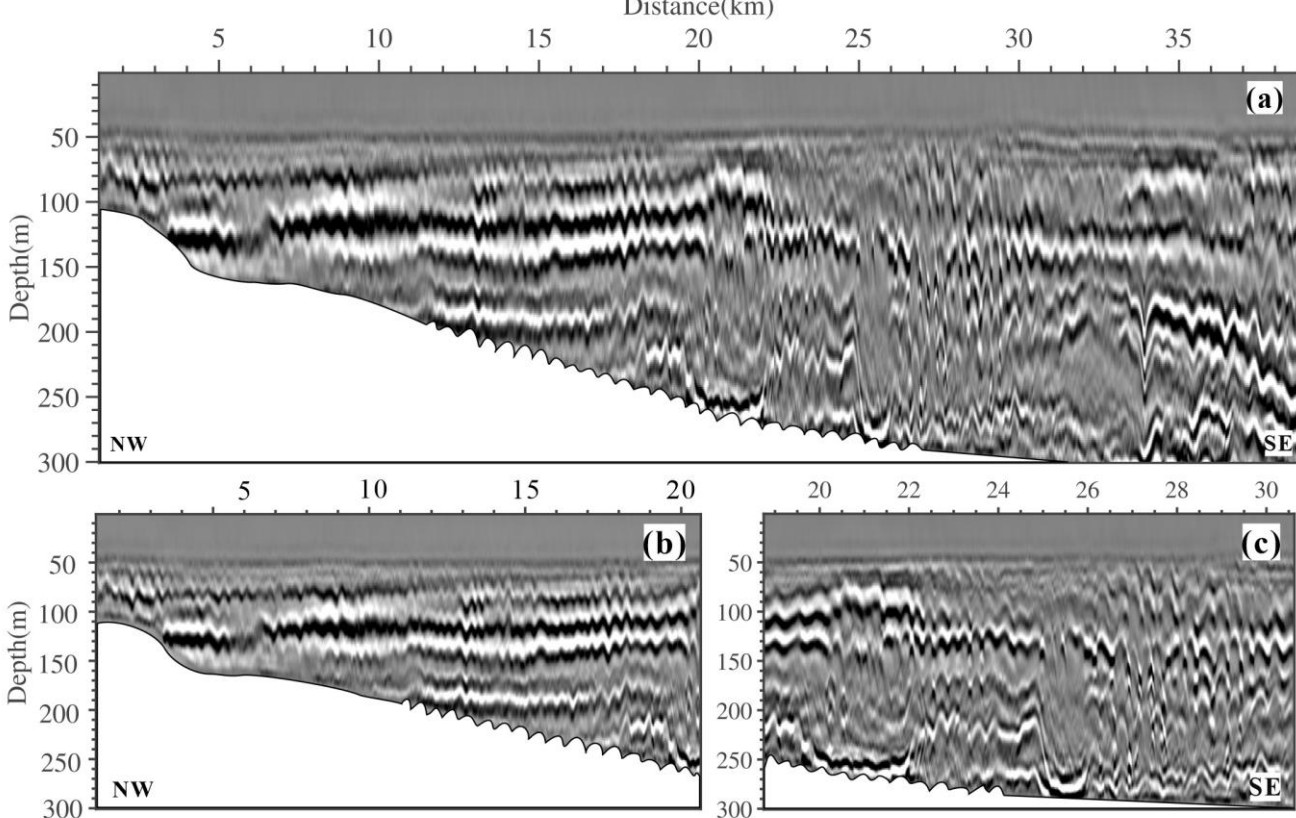

**Figure 6.** (a) Seismic profile of the 1.25-38.75 km section on line 25; (b) High-frequency internal waves; (c) High-mode internal waves and
K-H billows.

## 3.2 High-frequency internal waves

The depth of the thermocline on the shelf can shallow or deepen due to the influence of the seabed topography. Generally, during the shoaling process of internal waves or internal solitary waves, as the seabed becomes shallower, the depth of the pycnocline decreases, the density difference between the upper and lower layers diminishes, and the phase velocity of the wave
reduces. This shoaling process is often accompanied by the fission of descending waves (Djordjevic and Redekopp, 1978; Liu et al., 1998; Zheng et al., 2001). XBT data collected in September 2009, totaling nine, with their alignment roughly in the same direction as the distribution of line 25, allow a rough determination of the variation in thermocline depth with changes in water depth (Fig. 1a, Fig. 7). We consider the mid-depth between two inflection points on the temperature profile (where there is a significant temperature change) as the depth of the thermocline. According to Table 1, it can be seen that as the seabed





topography becomes shallower, the depth of the thermocline decreases from 127 m to 68 m.

In Fig. 6b, the high-frequency internal waves (packets) produced during the shoaling of internal solitary waves are mainly distributed within a depth range of 79-184 m, with an average amplitude of 7-9 m and a maximum amplitude of about 13 m. The half-height width ranges from 154-240 m, as detailed in Table 2. This is similar to the size of high-frequency internal waves observed by Bai et al. (2019). The reflection layer is near the pycnocline, that is, at a depth of nearly a hundred meters,

where there is high-frequency oscillation and good horizontal continuity. Near the seabed, due to strong interactions between internal waves and the terrain, the reflection layer distorts and gradually loses its horizontal continuity (horizontal distance 18-20 km, depth 200-250 m). Near the seabed, we can also see submarine dunes with heights of 1.5-20 m and wavelengths of 55-510 m, belonging to large or giant dunes (Ashley, 1990). This may be the result of strong near-seabed currents induced by internal solitary waves affecting sediment transport and accumulation.

Gong et al. (2021a) conducted a detailed statistical study on the amplitude, wavelength, and other parameters of symmetrical internal solitary waves in this grid (Tables 1 and 2). These waves maintain their original shape during propagation in deeper water (>350 m) and have not yet strongly interacted with the seabed, so they can be considered as initial waves. The seabed slope corresponding to line 25 is known to be $S$=1.03° (0.018) (Fig. 1b). Calculations show that the range of wave steepness $a/L_w$ is 0.01-0.36, and the range of $\xi_{in}$ is 0.03-0.17. Therefore, internal solitary waves usually undergo fission under

conditions of gentle slope ($S\leq0.05$) and low internal Iribarren number, producing a series of elevation waves (Orr and Mignerey, 2003; Sinnett et al., 2022; Zheng et al., 2001). The high-frequency internal waves seen in Fig. 6b might also be the products of the fission process (Bai et al., 2013; Bai et al., 2019). Additionally, $a/L_w$ is generally proportional to the Froude number, used to describe the degree of nonlinearity of internal waves. A ratio greater than 1 would result in seawater overturning (Masunaga et al., 2019), while here the value is less than 1.

Given the known range of initial wave amplitude $a_m$ and $H_m$ (=127 m, which is the thermocline depth of the initial wave), according to formula 2, the depth range $H_b$ where internal wave breaking occurs is 136-231 m, corresponding to a horizontal range of 5-15 km in the profile. However, we did not observe any breaking events in Fig. 6, and the most likely reason is that the seabed slope is gentle enough that none of the other three types of breaking mentioned earlier occurred. Therefore, the seabed slope is one of the key factors determining the shoaling and breaking of internal waves, consistent with other research

findings (Aghsaee et al., 2010; Ghassemi et al., 2021; Masunaga et al., 2019; Terletska et al., 2020).



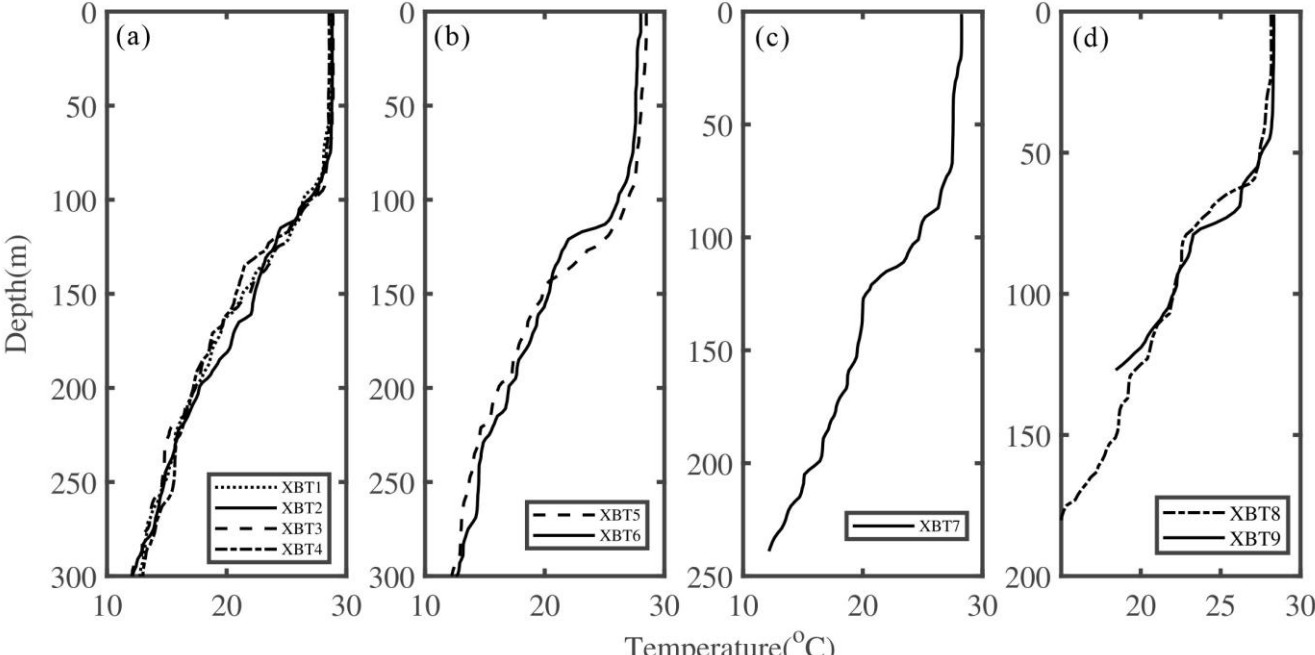

**Figure 7.** Temperature profiles from XBT. The distribution of XBT stations is listed in Fig. 1b.

**Table 1.** Changes in Thermocline Depth.

| XBT# | Thermocline (m) |
|---|---|
| XBT1/XBT2/XBT3/XBT4 | 127(91-163) |
| XBT5/XBT6 | 119(111-127) |
| XBT7 | 96(71-121) |
| XBT8/XBT9 | 68(57-79) |


**Table 2.** Statistics of High-Frequency Internal Waveforms.

| HIWs# | $Num$ | $\bar{A}$ (m) | $A_0$ (m) | $\bar{L}$ (m) | $\bar{D}$ (m) |
|---|---|---|---|---|---|
| 1 | 39 | 8.2 | 12.8 | 205.4 | 79.1 |
| 2 | 26 | 9.3 | 12.3 | 213.4 | 87.9 |
| 3 | 7 | 6.9 | 11.7 | 201.9 | 104.2 |



| 4 | 47 | 6.7 | 11.5 | 212.1 | 120.6 |
| 5 | 28 | 7.9 | 12.0 | 240.0 | 144.4 |
| 6 | 29 | 7.3 | 10.2 | 191.5 | 173.6 |
| 7 | 24 | 8.4 | 11.4 | 153.9 | 184.0 |

HIWs#, the number of high-frequency internal waves (packets); Num, the number of elevation waves in the packet; $\bar{A}$, average amplitude; $A_0$, maximum amplitude in the packet; $\bar{L}$, average half-height width; $\bar{D}$, average depth of the packet location.

**3.3 High-mode nonlinear internal waves**

Figure 8 is an enlarged view of the 19-27 km section in Fig. 6. We can see that the upper boundary of these two "convex" structure reflectors is convex upwards, while the lower boundary is concave downwards (Fig. 8). Compared to the strong reflections from the upper and lower boundaries, the interior of the reflectors is transparent or weakly reflective, with the central depth around 180 m and a thickness of about 150 m, indicating a well-mixed water layer. Because the waveforms are similar to the convex mode-2 internal solitary waves described by Yang et al. (2010), we preliminarily infer that these are convex mode-2 internal waves, sequentially numbered from left to right as HISW1 and HISW2.

Near a horizontal distance of 20 km, the upper boundary layer of the wavefront no longer remains horizontal but gradually becomes convex upwards with about a 70 m displacement, while the lower boundary becomes concave downwards with about a 50 m displacement. In contrast, the upper and lower boundaries at the wave rear (near 22 km) have opposite displacements, forming a high-mode nonlinear internal wave, i.e., HISW1. Due to the presence of internal tidal bores (indicated by the blue dashed line in Fig. 8b), the seawater at the wave rear gradually returns to its original stratification (22-24 km), and high-frequency internal waves reappear. However, about 2 km behind HISW1, another similar-shaped mode-2 internal solitary wave (HISW2) appears. The upper boundary of this wave is more diffuse, and the lower boundary layer is concave down to near a depth of 290 m with about a 100 m displacement. The wave rear (near 26 km) may be affected by shear instability, showing signs of waveform breaking. Additionally, unlike the former, the reflection layer behind HISW2 does not directly return to the original seawater stratification but becomes disordered due to instability, resulting in the appearance of K-H billows and the seawater being in a state of strong turbulence.

It is worth noting that, according to our calculation, we found that the apparent wave width range of these two high-mode nonlinear internal waves is 1.5-2 km, with an amplitude range of 50-100 m, which is clearly larger than the size of the mode-2 internal solitary waves found by Yang et al. (2009) in the South China Sea shelf and slope areas (with amplitudes of 20-30 m and an average time scale of 6.9-8 minutes). They might belong to exceptionally large-amplitude internal solitary waves (Brandt and Shipley, 2014).





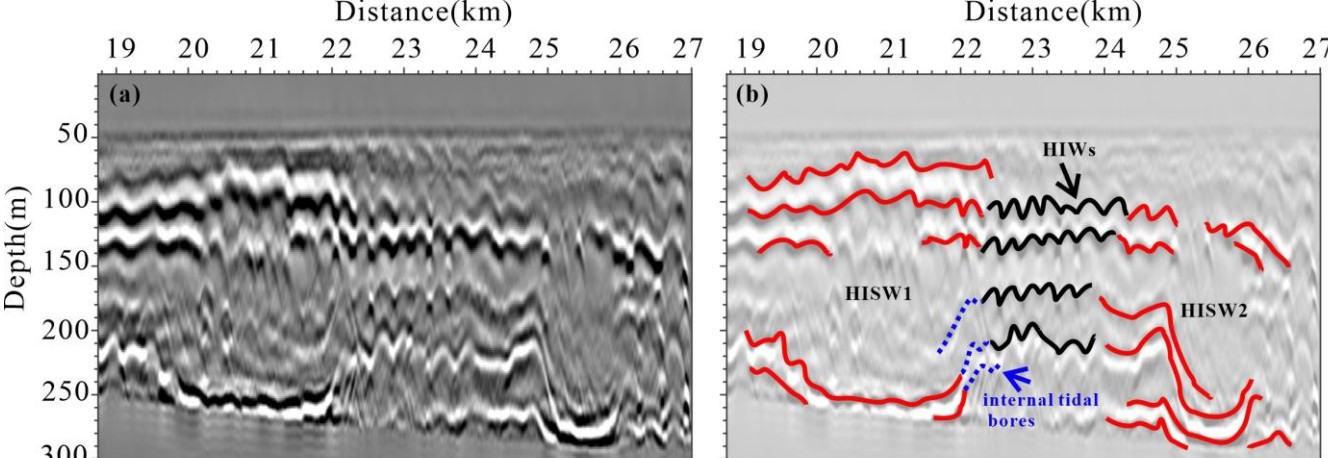

**Figure 8.** High-mode nonlinear internal waves. (a) Enlarged view of the 19-27 km section in Fig. 6; (b) Interpretative diagram of the reflection layers in the seismic profile, where red solid lines represent HISWs, black solid lines represent high-frequency internal waves, and blue dashed lines indicate internal tidal bores.

## 3.4 Shear instability

We know that in stratified shear flows, when the shear flow is strong enough or the stratification is weak enough, it will cause the Richardson number to fall below the critical value (0.25), thereby leading to the occurrence of shear instability (Smyth and Moum, 2012), which is divided into Kelvin-Helmholtz (KH) instability and Holmboe (HB) instability, with the former being the most common (Tedford et al., 2009). In Fig. 6b, both of the mode-2 internal solitary waves exhibit instability in their rear wings, especially HISW2, whose right-side waveform is not complete, and behind which very clear K-H billows are observed.

As shown in Fig. 9, using seismic data, we can very clearly see the true form of K-H billows, including amplitude, apparent wavelength, and distribution depth, etc. The billows are distributed in the range of 26.5-30 km, mostly inclined inverted "S" shapes, i.e., braid structures, and do not exhibit a complete braid-core structure. Table 3 details some of the clearer K-H billows in the profile. The amplitudes of these billows range between 26-45 m, with apparent wavelengths between 212-322 m. When less than 28.2 km, the waveform of the billows is asymmetric, and the shear instability is very strong, while at greater than 28.2 km, the waveform becomes more symmetric, with also strong shear instability. However, near 30 km the billows almost disappear, and the reflection layer starts to become continuous again, indicating that the shear instability weakens (Fig. 9a, 9b)

Due to the insufficient resolution of instruments such as CTD, XBT, and current meters, small-scale turbulent structures are often overlooked, resulting in estimates of the Richardson number far higher than the actual value (Moum et al., 2003; Tu et al., 2020). However, using the method mentioned in Sect 2.3, we can extract the waveform information of K-H billows from seismic profiles and estimate the values of Aspect ratio and $Ri_o$, thereby roughly assessing the instability of seawater. Through calculation, the average Aspect ratio is 0.13, and the $Ri_o$ averages 0.2, indicating conditions favorable for the occurrence of



instability (Table 3).

Tu et al. (2022) estimated the turbulence dissipation rate ε caused by K-H billows using waveform information extracted from acoustic profiles:

$$\varepsilon = C^{-2} h_{es}^2 N^3,$$  (6)

where $C \approx 12.5$, $h_{es}$ is the amplitude (wave height), and $N$ is the buoyancy frequency. Here, we can apply this parameterization scheme to seismic data to estimate the turbulence dissipation rate. The curve of buoyancy frequency $N$ as it varies with depth is shown in Fig. 5d. The calculated average dissipation rate $\varepsilon = 10^{-6.7 \pm 0.1} \, m^2 s^{-3}$, indicating that shear instability plays an important role in the energy dissipation of internal solitary waves and is often a key link in the transfer of internal wave energy to small-scale turbulence.

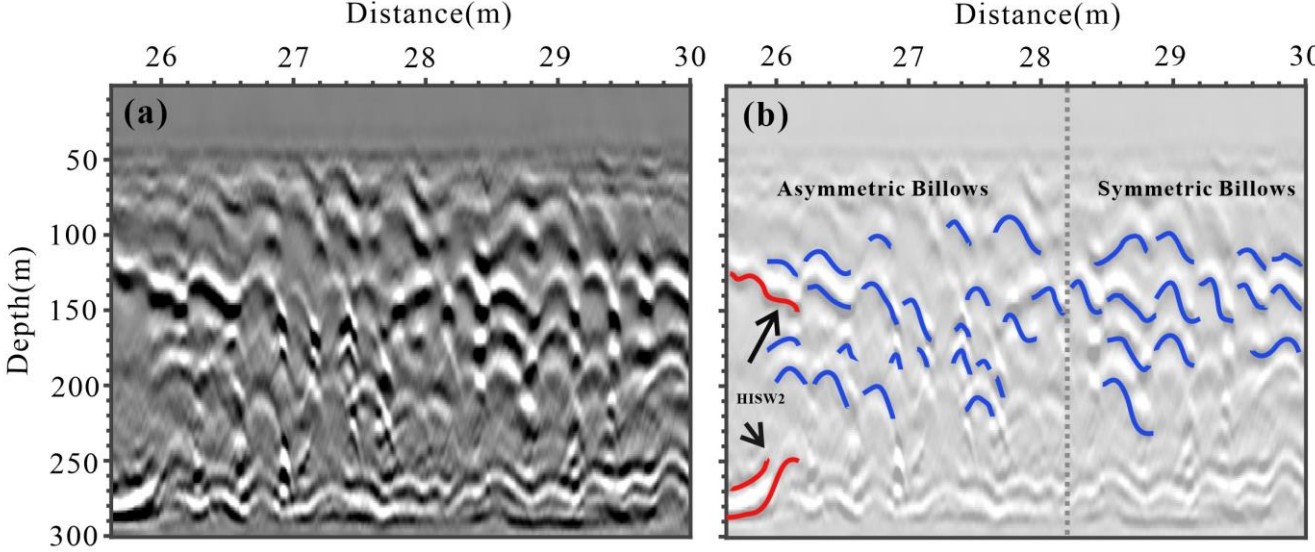

**Figure 9.** Structure of K-H billows. (a) Enlarged view of the section containing K-H billows; (b) Interpretative diagram of the reflection layers in the seismic profile, where blue solid lines represent K-H billows, grey dashed lines are used to distinguish the symmetry of the waveforms with billows on the left being asymmetric and most on the right being symmetric, and red solid lines represent the rear wing of HISW2, whose waveform is not very complete due to instability.

**Table 3.** Statistics of K-H Billow Waveforms.

| Billows# | Amplitude (m) | Wavelength (m) | Distance (m) | Depth (m) | Symmetry | Aspect Ratio | $Ri_o$ |
|----------|---------------|----------------|--------------|-----------|----------|--------------|--------|
| 1 | 31.6±0.8 | 212.6±2.9 | 29.6 | 128.5 | S | 0.15 | 0.19 |
| 2 | 41.8±0.1 | 241.0±8.7 | 29.4 | 130.3 | S | 0.17 | 0.18 |





| 3 | 33.3±0.3 | 258.6±3.6 | 29.1 | 135.3 | S | 0.13 | 0.20 |
| 4 | 31.7±0.3 | 328.1±7.7 | 28.7 | 139.9 | S | 0.10 | 0.21 |
| 5 | 34.0±0.7 | 221.0±14.2 | 28.4 | 134.7 | A | 0.15 | 0.19 |
| 6 | 34.7±0.3 | 318.9±24.8 | 28.1 | 139.2 | A | 0.11 | 0.21 |
| 7 | 32.8±0.3 | 284.2±12.3 | 28.9 | 172.4 | S | 0.12 | 0.20 |
| 8 | 33.9±1.7 | 322.1±4.1 | 28.7 | 166.9 | S | 0.11 | 0.21 |
| 9 | 26.2±0.5 | 236.9±0.5 | 28.3 | 157.0 | A | 0.11 | 0.21 |
| 10 | 44.9±1.4 | 373.1±1.4 | 28.7 | 200.7 | A | 0.13 | 0.20 |
| 11 | 28.8±1.4 | 234.5±12.3 | 29.1 | 195.6 | A | 0.12 | 0.20 |
| 12 | 43.2±2.5 | 303.2±29.6 | 27.7 | 211.6 | A | 0.14 | 0.19 |

Billows#, the number of K-H billows; Amplitude, the amplitude (wave height) of the billows; Wavelength, the wavelength of the billows; Distance, the horizontal distance where the billows are located; Depth, the depth of the seawater where the billows are located; Symmetry, the symmetry of instability, S=Symmetric, A=Asymmetric.

**3.5 Diapycnal diffusivity maps**

We used the method mentioned in Sect 2.4 to interpolate and smooth the average dissipation rate (diapycnal mixing) calculated for each window with the values from the surrounding windows. Finally, overlaying the results on the seismic image, we obtain Fig. 10a (10b). This not only makes it easier to grasp the spatial distribution of the calculation results but also to understand the correspondence between the results and the seismic image. Figure 11 is a histogram of the dissipation rates (diapycnal mixing) over different horizontal ranges.

According to Fig. 10 and Fig. 11, we can see that the diapycnal mixing for the turbulent segment is calculated as $10^{-4.10\pm0.09}\ m^2s^{-1}$ (-4.10 is the average and 0.09 is the standard deviation), which is an order of magnitude greater than the average for the open ocean ($10^{-5}$). In the upper layer of seawater (above 200 m), especially near the thermocline, the dissipation rate (diapycnal mixing) is high, while below 200 m, the dissipation rate (diapycnal mixing) decreases with increasing depth. However, near the seabed, that is, in the slope and shelf areas, the dissipation rate (diapycnal mixing) is also high. The

calculated diapycnal mixing for the 1.25-38.75 km range is $10^{-3.84\pm0.11}\ m^2s^{-1}$, which is 3.5 times greater than the calculation for the 62.5-88.75 km range ($10^{-4.39}\ m^2s^{-1}$). As we discussed earlier, a series of shoaling events occurred within the 1.25-38.75 km range: the formation of high-frequency internal waves due to internal wave shoaling, the formation of high-mode nonlinear waves and the induced shear instability due to strong interaction with the seabed topography. These all cause stronger energy dissipation, ultimately leading to the breaking of oceanic internal waves and facilitating the downscaling



transfer of energy.

**Figure 10.** Distribution map of mixing parameters. (a) Spatial distribution of turbulence dissipation rate; (b) Spatial distribution of diapycnal mixing, where the blue arrow represents the horizontal range of 1.25-38.75 km, the red arrow represents the horizontal range of 62.5-88.75 km, and the black arrow represents the horizontal range of 1.25-88.75 km.





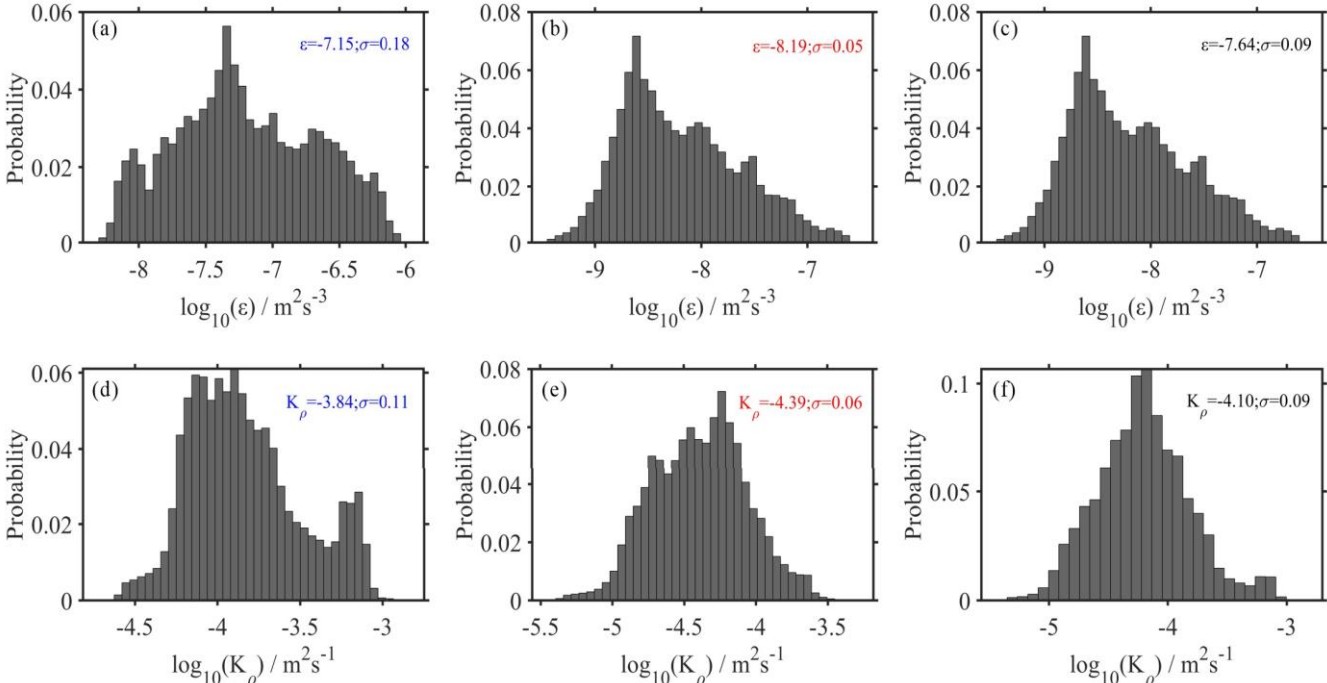


**Figure 11.** (a) and (d) are distribution histograms of the dissipation rate and diapycnal mixing, respectively, for the range indicated by the blue arrow (1.25-38.75 km) in Fig. 11; (b) and (e) are distribution histograms of the dissipation rate and diapycnal mixing, respectively, for the range indicated by the red arrow (62.5-88.75 km) in Fig. 11; (c) and (f) are distribution histograms of the dissipation rate and diapycnal mixing, respectively, for the range indicated by the black arrow (1.25-88.75 km) in Fig. 11.

**3.6 Error analysis**

Generally speaking, there is a certain error between the fitted spectrum obtained by the least squares method and the actual spectrum, which we represent here using the standard deviation $\sigma$. Therefore, we can obtain the upper and lower limits of the turbulence dissipation rate (diapycnal mixing) (Fig. 12).

Additionally, the selection of different parameter values in formulas 3 and 4 will also result in errors in the outcome. We 335 typically take the 95% confidence interval of the average buoyancy frequency (Fig. 5d), so the error in the diapycnal mixing in the log domain is about 0.02 $m^2 s^{-1}$. The range of the mixing efficiency $\Gamma$ is 0.1-0.4 (Mashayek et al., 2017), and here we take 0.2, causing an error in the diapycnal mixing in the log domain of about 0.15 $m^2 s^{-1}$. The range of $C_T$ values is 0.3-0.5 (Sreenivasan, 1996), and here we take 0.4, resulting in an error in the diapycnal mixing of about 0.14-0.17 $m^2 s^{-1}$ in the log domain.








**Figure 12.** (a) ε-σ; (b) ε+σ; (c) $K_\rho$-σ; (d) $K_\rho$+σ

## 4 Discussions

### 4.1 Mode-2 internal solitary waves or shoaling bores?

In Sect 3.3, we only preliminarily inferred that the two "convex" structure reflectors in Fig. 8 are exceptionally large-amplitude
mode-2 internal solitary waves based on waveform similarity, which still requires further discussion. Utilizing field
observations and numerical simulations, Scotti et al. (2008) found that strong nonlinear high-frequency internal waves
interacting strongly with shoaling terrain in Massachusetts Bay can result in "Seabed Collision Events" that is, the waveform
of internal waves will change into different shapes during shoaling. Specifically, they found that under conditions of gentle
slope and moderate amplitude, high-frequency internal waves in the slope and shelf areas would undergo collision events with
strong energy, producing waveforms very similar to those observed in Fig. 8, and also accompanied by hydraulic jump.
However, they did not explicitly state that these were mode-2 internal solitary waves; the appearance of these complex wave
packets is just one manifestation of internal wave deformation during these events. Therefore, we infer that the two convex
structure reflectors might be formed from strong interactions between high-frequency internal waves and the terrain, but we
still cannot confirm whether they are mode-2 internal solitary waves.

Shoaling internal waves usually form bores with bottom-enclosed structures carrying cold water masses that continue to
propagate shoreward after breaking (Jones et al., 2020). The strength and structure of bores are influenced by the background
stratification and the depth of the thermocline (if $(A+z_{th})/h > 1/2$, shoaling internal waves will form bores, where $A$ is the
amplitude of the internal solitary wave, $z_{th}$ is the thermocline depth, and $h$ is the water depth) (Scotti et al., 2008; Sinnett et al.,
2022; Walter et al., 2014). Bores typically begin to dominate the structure of internal waves near depths less than 50 m,
becoming a primary feature of the wave during the shoaling process (McSweeney et al., 2020). However, the two complex
wave packets in this paper are located at a depth of 300 m. Therefore, we believe that although the structure of the two wave
packets is somewhat similar to that of bores, their distribution depth range does not align with the typical distribution depth of
common bores, which are often in shallower regions at the run-up phase of shoaling internal waves. Thus, we prefer to interpret
the two "convex" structure reflectors as high-mode (mode-2) internal solitary waves.

### 4.2 Dissipation mechanism of internal solitary waves in the South China Sea

When energy from large and medium-scale motions is transferred to small-scale turbulence, internal waves often play a very
important role (Liu et al., 2022). The shoaling evolution and dissipation of internal solitary waves in shelf and slope areas are
key links in driving turbulent mixing. Previous extensive studies have shown that internal solitary waves enhance the mixing
of seawater during shoaling (Gong et al., 2021b; Moum et al., 2007). Lamb (2014) summarized four scenarios in which internal
solitary wave shoaling enhances seawater mixing: 1) Vertical shear flow caused by internal solitary waves leads to shear
instability in seawater, usually occurring in areas where Ri < 0.25; 2) Convective instability occurs when the velocity of the





water mass exceeds the phase velocity, often accompanied by enclosed vortex cores, within which the water is unstable and in a turbulent state; 3) Global instability near the seabed boundary layer causes sediment resuspension and transport; 4) Direct breaking occurs when encountering steep terrain during the shoaling process.

The slope and shelf areas of the South China Sea are usually quite gentle, and internal solitary waves do not directly break but instead undergo fission under dissipative effects. The trailing part of the leading wave generates a series of elevation waves, and a significant portion of the energy of the internal solitary wave continues to dissipate into the seawater through these trailing waves. This process also causes instability in the seawater, leading to strong turbulent mixing (Moum et al., 2003; Moum et al., 2007). In this paper, through calculations, we find that the high-frequency internal waves generated during the

shoaling of internal waves can also produce strong turbulent dissipation and mixing, on average causing diapycnal mixing of the order of $10^{-4}$ (Fig. 10, 11d). This indicates that high-frequency internal waves are also a way for internal solitary waves to dissipate energy. Near the high-mode nonlinear internal waves and K-H billows generated during shoaling, one can clearly see red patches (Fig. 10a, 10b), with a slightly deeper color than the area where high-frequency internal waves develop, indicating that this is a high mixing area of the entire profile, which can even reach the order of $10^{-3}$ (Fig. 11d). However, in the slope

area far from the high-frequency internal waves and K-H billows, the diapycnal mixing is slightly lower, with the diapycnal mixing in the shelf area being about 3.5 times that of the slope area. As analyzed in Sect 3.3 and 3.4, under the influence of various conditions such as topography, stratification, and background flow, the dispersion effect gradually becomes dominant. The HISWs generated during shoaling undergo further deformation, with the waveforms gradually widening and losing their original symmetry, significantly increasing internal wave dissipation. The development of shear instability near the thermocline

causes the internal waves to break, leading to intense turbulence. Therefore, the fission of internal solitary waves and the accompanying shear instability are important pathways for the dissipation of energy of internal solitary waves in the slope and shelf areas of the South China Sea (Zhang et al., 2023), and the high-frequency internal waves generated during shoaling may also play an indispensable role in the process of energy dissipation of internal solitary waves (Bai et al., 2019; Liu et al., 2022).

**5 Conclusions**

In the seismic section, we identified high-frequency internal waves, mode-2 internal waves, and shear instability in the ocean. We believe that the strong nonlinear high-frequency internal wave packets are the result of the shoaling, primarily distributed within a depth range of 79-184 m, with amplitudes of $O$ (10) m and a half-height width range of 154-240 m. Fission usually applies to situations with gentle slopes and low $\xi_{in}$. Calculations show that the seabed slope corresponding to line 25 is $S$=0.018 ($\leqslant$0.05), and $\xi_{in}$ ranges from 0.03-0.17. Behind the high-frequency internal waves, there are two mode-2 internal

solitary waves with larger amplitude and wave width, which may have formed due to strong interaction between internal waves and the terrain. In the seismic profile, we observed that both mode-2 internal solitary waves exhibit instability in their rear wings, especially HISW2, whose tail waveform is incomplete and has very clear K-H billows, a rarity observed in seismic sections. The amplitudes of these billows range between 26-45 m, with apparent wavelengths between 212-322 m, the average

Aspect ratio is 0.13, and the average $Ri_o$ is 0.2. Based on the parameterization scheme, the average dissipation rate ε caused by shear instability is calculated to be $10^{-6.7\pm0.1}\ m^2 s^{-3}$, indicating that shear instability plays an important role in the energy dissipation of internal solitary waves.

We used seismic data to estimate the mixing parameters of seawater and found that the diapycnal mixing caused by these shoaling events in the shelf area is 3.5 times greater than that in the slope area. The shoaling process of internal solitary waves in the slope and shelf areas is essentially a process of energy dissipation of the internal solitary waves, continuously enhancing the mixing of seawater. This process, accompanied by a series of shoaling events such as high-frequency internal waves, high-mode nonlinear internal waves, and shear instability in seawater, is the most direct manifestation. They eventually lead to the breaking of internal waves, facilitating the downscaling transfer of energy.

*Code and data availability.* The seismic data was processed using Seismic Unix developed by the Center for Wave Phenomena (CWP) at the Colorado School of Mines. XBT data comes from the National Oceanic and Atmospheric Administration's World Ocean Database 2018 (WOD18, https://www.ncei.noaa.gov/OCL/). CTD data is sourced from the Climate and Ocean Project and Carbon Hydrographic Data Office (CCHDO, https://cchdo.ucsd.edu/).

*Author contribution.* LM completed this paper under the guidance of Professor HS and YG. LM processed and analyzed the data, and drafted the manuscript. SY, KZ, and ML discussed the results and revised the manuscript.

*Competing interests.* The authors declare that they have no conflict of interest.

*Acknowledgements.* The seismic data are owned by the Guangzhou Marine Geological Survey (GMGS). Thanks to the GMGS for providing two-dimensional seismic data.

*Financial support.* This work was funded by the National Natural Science Foundation of China (Grants 42176061, 41976048).

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
