# Peer review of "Energy Transfer from Internal Solitary Waves to Turbulence via"

_EGUsphere, 2024_

## Referee Comment (RC4)

Report on the paper
**"High-frequency Internal Waves, High-mode Nonlinear Waves and KH
Billows on the South China Sea's Shelf Revealed by Marine Seismic Observation"**
by Linghan Meng et al.

**Summary**
This paper aims at identifying the structure of nonlinear motions (internal solitary waves) responsible for mixing on the shelf and slope of the South China Sea using marine seismic observations. As clearly stated in the introduction of the paper, the structure and formation mechanisms of these motions have been already identified in numerical simulations and the challenge here is to perform a similar analysis from field measurements using seismic data.

**Comments and recommendation**
The analysis performed by the authors is interesting, as it allows to quantify the mixing induced by internal solitary waves and related motions from seismic observations. The bibliography is abundant and it is clearly demonstrated how the work in the paper differs from, and goes beyond, previous works published in the literature. My main comments relate to the writing of the paper.  The conclusion in this respect is poorly written and effort should be made to summarize the novelty of the results with respect to the literature. The paper also needs to be tightened. At many instances indeed, adjectives such as "strong" (e.g. l. 25) "relatively small" (l. 191), "large and medium-scale" (l. 365), etc. are used. These terms should be made quantitative or the authors should say strong/small etc. with respect to what.
**To my opinion this paper deserves to be published after these comments, and the other ones below, have been taken into account in a revised version**

**Other comments**

**1. Introduction**
l. 28: explain what "shoaling" means
l. 34: what do you mean by "critical depth"?
l. 35: "reversing polarity to form elevated waves": this is not clear, please explain better (or refer to a sketch in a published paper)
l. 36: "dispersion continues to form elevated waves": hard to understand if "elevated waves" is not explained.
l. 45: references are needed after "through numerical simulations".
l. 50: "we attempt to use" -> change into: we use
l. 51: "we were fortunate to observe" -> change into: we were able to observe.

**2.1 Seismic data acquisition and processing**
This is a very technical section and one reference (at least) is requested, for instance at the end of the second paragraph, on l. 65.
l. 59: "red solid line" -> black solid line, actually.

**Caption of Figure 1**
The caption should be understandable by the reader, which is not the case here. Indeed:

- "the line 25" should be explained or a reference to the line of the paper where this is explained should be provided.
- the difference between the "magenta" and the "red" colors is not easy to see -> use another colour for magenta (orange for instance).
- what does mean "calculated to have a seabed slope": is it just the calculation of the slope which yields this value of $\gamma$?
- About this parameter $\gamma$: there is a confusion between the angle and the slope (equal to the tangent of the angle) which are both denoted by $\gamma$. Thus "seabed slope $\gamma$ =1.08° (0.018)" should be replaced by "seabed angle 1.08° (associated with the slope $\gamma$ =0.018)".

**Caption of Figure 2**
Which quantity is displayed in each panel? Is it the isopycnal displacement? This should be said.
Also, avoid writing "F-K" spectrum and write explicitely "frequency-horizontal wavenumber spectrum".

**Section 4.2**
l. 373: what do you mean by "global instability"?
l. 381 and 384: "diapycnal mixing of the order of $10^{-4}$", "the order of $10^{-3}$": add physical unit.
l. 388: "HISW" has not been defined previously.

**Conclusion**
As indicated in the first general comment above, the conclusion is poorly written and should be rephrased following my comment.

---

## Author Comment (AC1)

**Responses to Reviewer 1**

The authors (AC) thank Reviewer 1 (RC1) for the valuable comments and good suggestions that greatly helped to improve our manuscript. Below are our itemized.

**RC1 Summary:** The paper discusses seismic observations of shoaling internal tides and waves, and describes features the authors identify in the data. the seismic observational approach is not new; its relevance and capabilities in relation to internal flows is not discussed; "feature" identification is superficial, based on the authors "impressions", rather than rigorous; we have no measure of the validity of these estimates. What exactly is the novel contribution of the manuscript?

**AC** Thank you for your valuable comments. We have carefully considered your comments and have made major revisions to the manuscript to address your concerns. Specifically, we have adjusted the content and structure of the manuscript to enhance its clarity and rigor. The corresponding updates are highlighted in red in the revised version. Below are our itemized.

**RC1-1** It is hard to understand what the contribution of this manuscript is. The authors make it clear that neither the observational methods (seismic method) nor the phenomena identified in the data are new (see the Introduction and Data and methods sections). The remaining potentially valuable contributions might be: the data set itself, and possible new phenomena the data set reveals.

**AC** Thanks for your valuable comment. We apologize for not explaining the contribution of this article clearly. We have carefully considered your comments and have made major revisions to the manuscript to address your concerns, using red colored texts.

In the slope-shelf regions of the South China Sea, where the topography is relatively gentle, internal solitary waves (ISWs) frequently appear as wave packets through fission, as confirmed by satellite image (e.g., Zhao et al., 2003). During this process, the energy of ISWs continuously dissipates into the seawater, enhancing turbulent mixing. Ultimately, these shoaling waves break due to shear instability or convective instability, generating turbulence and achieving the downscale transfer of ocean energy (Lamb et al., 2014). This represents our most common understanding of the dissipation mechanisms of ISWs.

In this manuscript, we observed HIWs resulting from the fission of ISWs in the northern South China Sea using seismic method, as confirmed by MODIS data. We use seismic data to estimate diapycnal mixing, and we found that the HIWs can enhance diapycnal mixing, averaging $10^{-4}$ m$^2$s$^{-1}$. The maximum mixing is up to $10^{-3}$ m$^2$s$^{-1}$, and it is associated with the breaking of IWs caused by the strong shear. Our study provides new evidence of the energy transfer mechanisms from shoaling ISWs to turbulence through HIWs, contributing to a deeper understanding of the energy cascade processes in this region. The observations and numerical simulations by Bai et al. (2013, 2019) support this point. And they also think the fission of ISWs into HIWs on the continental shelf is an important pathway for tidal energy cascade.

**RC1-2** The data set seems to be confined to the gray map shown in figure 2 (the "after" panel). Perhaps this data set is uniquely difficult to obtain and deserves a paper in itself. It seems doubtful (method not new). However, because this reviewer is not familiar with the seismic observational techniques, advantages and disadvantages, the quality of the data and its reliability will not be discussed here.

**AC** Thanks for your valuable comment. The seismic method, that is Seismic oceanography, has

provided us with a novel high-resolution method to image the thermohaline fine structure in the water column (Holbrook., 2003). Similar to high-frequency acoustic data, but this method uses low-frequency (about tens to hundreds of Hz) sound waves to image the fine structure in the ocean. The image formed by the reflection seismic method in the water column is mainly the result of the convolution of the seawater temperature gradient and the seismic source wavelet (Ruddick et al., 2009; Sallarès et al., 2009). In order to better illustrate what seismic images represent, Ruddick et al. (2009)'s Figure 1 (Figure R1 in this response letter) is cited here. It compares the conventional contour plot of temperature from a CTD with the seismic image.

Due to its advantages of fast acquisition, high horizontal resolution, and full-depth water column imaging, it has been widely proven to be an effective method for capturing ocean dynamic phenomena covering a multi-scale range from fine scale $O$ (10 m) to mesoscale $O$ (100 km) (Song et al., 2021a). Many ocean features are captured and imaged through this high-resolution seismic images, such as ocean currents (e.g., Tsuji et al., 2005), eddies (e.g., Yang et al., 2022), fronts (e.g., Gunn et al., 2021), internal solitary waves (e.g., Bai et al., 2017; Geng et al., 2019; Gong et al., 2021a; Song et al., 2021b; Tang et al., 2014, 2018), thermohaline staircases (e.g., Fer et al., 2010), and turbulence (e.g., Sheen et al., 2009).

We have added detailed descriptions about seismic oceanography method in the revised manuscript, using red colored texts (line 58-72).

[Figure]

Figure R1 (a) Conventional contour plot of temperature from a CTD transect of Meddy Sharon. (b) Seismic oceanography image

**RC1-3** The observations perhaps reveal something new and valuable, but after reading the manuscript, the features identified seem well known and understood. Perhaps just the fact they appear in that environment is a valuable piece of information? if so, a discussion is warranted (and in fact not found in the manuscript).

**AC** Thanks for your valuable comment. The observations are indeed well-known, but they often come from laboratories and numerical simulations, and the shoaling internal waves can rarely be clearly observed in situ. Of course, as mentioned in the manuscript, some scholars have also observed these phenomena using ADCP, high-frequency acoustic data, or moorings, but they may be limited to fixed-point observations or shallow water observations, and it is difficult to directly obtain spatial position distribution. And seismic data can help us solve these problems very well, as shown in Figure 5. We can clearly see the spatial distribution of these waves during the shoaling process, which helps us further understand the process.

In addition, there are many methods in the physical ocean that can be used to estimate turbulent mixing in the ocean, such as tracer release experiments, microstructure measurements, and parameterization schemes. However, these methods often have high difficulty, collection costs, and sparse data, which greatly limit the research on the spatiotemporal distribution and mechanism of ocean mixing. This is a significant obstacle as ocean mixing is often patchy in time and space. In the manuscript, we use seismic data to estimate diapycnal mixing, and we found that the HIWs can enhance diapycnal mixing, averaging $10^{-4}$ m$^2$s$^{-1}$. The results show a new energy cascade route from shoaling ISWs to turbulence, i.e., the fission of ISWs into HIWs, which improves our knowledge of ISW energy dissipation and their roles in improved mixing in the northern SCS.

**RC1-4** The discussion of the data set is particularly disappointing. The general narrative seems to be this: the authors identify some features in the data (marked in some figures by lines); declare that those features have a very specific meaning; and apply some well-known formulations to provide some quantitative discussion. The analysis of observational data should be approached with a lot of skepticism, and should involve careful cross-checking and verification. in this manuscript, the authors declare they see something and then weave a story around it. They see "high-frequency internal waves, mode-2 internal waves, and shear instability", and "believe that strong nonlinear high-frequency internal wave packets are the result of the shoaling. Do they see all this just from seismic reflections? This seems problematic. Is the seismic data enough to fully characterize the flows identified? It's hard to believe that it's so, because the dynamics of internal waves are described by complicated equations that involve quite a number of state variables. Why is seismic data enough to characterize these complicated states?

**AC** Thanks for your valuable comment. We have made major revisions to the manuscript and re-discussed the dataset. The more detailed version has been added to the revised manuscript, using red colored texts.

Due to the unique advantages of seismic methods in studying the vertical structure of water column (as mentioned above in RC2' response), this method has been widely used in the study of ISWs, and has achieved some significant research results. The relevant research is as follows: Tang et al. (2014) used pre-stack seismic data to calculate the apparent phase velocity of ISWs on the northeastern slope of the South China Sea, correlating this with hydrological and remote sensing data to analyze wave characteristics. Tang et al. (2015) optimized phase velocity calculations by grid search, selecting shot-receiver pairs with the best slope for the wave's leading edge. They calculated kinematic and dynamic parameters of the wave packets northeast of Dongsha Island and found that these waves align better with the KdV theory. Bai et al. (2015) used seismic methods to study the structural characteristics of ISWs in the Dongsha area, finding their energy in the low-wavenumber range to be about two orders of magnitude higher than the GM model spectrum. Tang et al. (2016) combined hydrological and reflection seismic data with the Markov chain Monte Carlo method to detail the fine structure of temperature and salinity in seawater, achieving detailed imaging of the ISWs. Bai et al. (2017) analyzed the polarity conversion process of shoaling ISWs in the northeastern South China Sea using reflection seismic data. Huang et al. (2018) combined fluid dynamics numerical simulations and reflection seismic forward modeling to study the geometric and structural characteristics of seawater layers during different stages of shoaling. Tang et al. (2018) used high-resolution reflection seismic imaging and simultaneous hydrological data to invert secondary ISWs on the northern shelf of the South China Sea, directly observing KH

instabilities. Geng et al. (2019) combined seismic data with KdV theoretical models to study the vertical structure of ISWs, finding that their maximum amplitude decreases logarithmically with depth. Gong et al. (2021a) used linear and nonlinear vertical structure theories to study ISWs in the northeastern South China Sea, concluding that nonlinearity, topography, and background flow influence their vertical structure. Sun et al. (2019), Fan et al. (2020), and Kuang et al. (2021) used marine seismic methods to study the structure and propagation of ISWs near the Strait of Gibraltar and the northeastern South China Sea. Gong et al. (2021b) captured three polarity-reversing ISWs near Dongsha Island using seismic sections, discovering that the mixing rate induced by these waves is two orders of magnitude higher than in the open ocean, with the diapycnal mixing rate of polarity-reversing waves being three times higher than non-reversing regions. Fan et al. (2021, 2022) used reflection seismic data to discover numerous shoaling mode-2 ISWs in Central America, analyzing their amplitude and propagation speed relationship and regional distribution. Song et al. (2021b) observed the shoaling process of ISWs through common-offset gathers pre-stack migration profiles, documenting changes in phase velocity, amplitude, wavelength, and slopes, further enhancing our understanding of ISWs and their shoaling processes. In a word, the seismic method has been widely recognized.

Secondly, to increase credibility, we added other datasets including remote sensing data and tidal models to Section 3.2 in the revised manuscript, using red colored texts (line 215-241). MODIS image (Figure 6, Figure RC2 in the response letter) revealed that ISWs from the Luzon Strait typically appeared as wave packets near Line 25 during the data collection period, with tidal models indicating these waves originated during neap tide.

There are two nonlinear waves (NIWs) with complex structures, and their shapes like bores (Figure 13 in the revised version). Given the complexity of the shape, we are not sure what they are exactly. Therefore, we place this part in the discussion section (Section 4.1, 4.2, line 376-413).

[Figure]

Figure R2 Satellite images and tidal model. (a) MODIS image collected by the Aqua satellite in the northern South China Sea on July 29, 2009; (b) MODIS image collected by the Terra satellite

in the northern South China Sea on July 30, 2009; (c) Tidal height time series near the Luzon Strait (20.6° N, 121.9° E) from July 14 to August 14, 2009. The red box indicates the time period used to infer the generation of ISWs.

**RC1-5** Which raises another question that is completely ignored by the authors: what is the place of seismic data among other observation techniques? In other words - what should or should not be inferred based on seismic observations? How much can we trust these features.

**AC** Thanks for your valuable comments. We apologize for the lack of a detailed explanation of seismic oceanography techniques in the manuscript. Seismic reflection sections provide very high-resolution images of oceanic features, both vertical and, in particular, horizontal direction, and complement conventional physical oceanography CTD/XBT data (e. g. Ruddick et al., 2009; Song et al., 2012b). These seismic images of the water column are primarily images of vertical temperature gradient convolved with the resolution scale of the seismic source wavelet, giving a resolution of typically ~10 m (Song et al., 2021a). Due to its advantages of fast acquisition, high horizontal resolution, and full-depth water column imaging, it has been widely proven to be an effective method for capturing ocean dynamic phenomena covering a multi-scale range from fine scale $O$ (10 m) to mesoscale $O$ (100 km). Therefore, mesoscale and submesoscale phenomena such as eddies, fronts, internal waves, thermohaline intrusions, and vortical modes can be directly observed via seismic techniques. The Table 1 (Table R1 in this response letter) and Figure 1 (Figure R3 in this response letter) in Song et al. (2021a) clearly explains these concerns.

we have added detailed descriptions of seismic observations (line 58-69): Seismic oceanography, initially proposed by Holbrook (2003), has provided us with a novel high-resolution method to image the thermohaline fine structure in the water column. The image formed by the reflection seismic method in the water column is mainly the result of the convolution of the seawater temperature gradient and the seismic source wavelet (Ruddick et al., 2009; Sallarès et al., 2009). Due to its advantages of fast acquisition, high horizontal resolution, and full-depth water column imaging, it has been widely proven to be an effective method for capturing ocean dynamic phenomena covering a multi-scale range from fine scale $O$ (10 m) to mesoscale $O$ (100 km) (Song et al., 2021a). Many ocean features are captured and imaged through this high-resolution seismic images, such as ocean currents (e.g., Tsuji et al., 2005), eddies (e.g., Yang et al., 2022), fronts (e.g., Gunn et al., 2021), internal solitary waves (e.g., Bai et al., 2017; Geng et al., 2019; Gong et al., 2021a; Song et al., 2021b; Tang et al., 2014 ; Tang et al., 2018), thermohaline staircases (e.g., Fer et al., 2010), and turbulence (e.g., Sheen et al., 2009). In recent years, this method has gradually become more quantitative, especially in estimating turbulent dissipation rates and diffusion (e.g., Dickinson et al., 2017; Gong et al., 2021b; Tang et al., 2021; Yang et al., 2023).

Table R1 after Song et al. (2021a). Comparison between conventional physical oceanography and seismic oceanography in the study of the seawater thermohaline structure.

| | Conventional physical oceanography | Seismic oceanography |
|---|---|---|
| **Measurement equipment** | CTD (Conductivity-Temperature-Depth), XBT (Expendable Bathy Thermograph System), XCTD (Expendable Conductivity-Temperature-Depth) | Multichannel seismic reflection method (airgun, hydrophones) |

| Horizontal resolution | 5–50 km | 6.25 m |
|---|---|---|
| Measuring time (as an example of 3 km water depth) | 3 h/CTD station | 4 s/CMP |
| Results | Temperature, salinity | Seawater structure, temperature, salinity, density, current velocity |
| Scope of study | large scale structure | meso scale and small scale structure (eddies, internal waves, fronts, etc.) |

[Figure]

Figure R3 Cascade of mechanical and thermal energy in the ocean from large forcing scales (top left), through intermediate scales and processes, and ultimately to molecular dissipation scales (a few mm, lower right). Mesoscale and Sub-mesoscale phenomena such as eddies, fronts, internal waves, thermohaline intrusions, and vortical modes can be directly observed via seismic techniques (From Figure 1 in Song et al., 2021a).

**RC1-6** And then, what are these identified features? What does the gray map in figure 2 represent? What do the lines drawn on the gray levels mean? this might be clear to people familiar with seismic data, but this manuscript is, as the title seems to say, about internal waves, not seismic data.

**AC** Thanks for your valuable comments. Reflectors in the seismic section are caused by variations in acoustic impedance (the product of density by sound speed), primarily controlled by changes in sound speed (90%-95%), which depend on the seawater temperature gradient (80%) (Ruddick et al., 2009; Sallarès et al., 2009). Therefore, the seismic section (the gray map) essentially represents a

high-resolution snapshot of the ocean's vertical temperature gradient, and these reflections (black-white lines) represent the isopycnals. We have added detailed descriptions of seismic observations (line 58-69).

**RC1-7** The Introduction section is a general discussion about the observational method and known internal waves phenomena. It does not identify a question that would give the manuscript a purpose. **AC** Thanks for your valuable comments. We have rewritten the Introduction section of the manuscript and clarified the research question and significance, using red colored texts (line 52-57): The energy dissipation mechanisms of IWs (ISWs) are complex and the ocean is a multi-scale coupled system with various dynamic processes. However, most current studies consider the ideal scenario where only IWs are present in the ocean's dynamic system, with few considering or only considering simplified dynamic environmental impacts. This results in an incomplete understanding of the shoaling processes and ISWs' dissipation mechanisms. Additionally, whether HIWs, another product of fission, significantly enhance energy dissipation and mixing remains a critical question for comprehending the formation and evolution of wave packets and their role in ocean mixing and energy cascades.

**RC1-8** The part of the "Data analysis and methods" section concerned with the discussion of seismic data processing is incomprehensible for someone that is not familiar with the technique. The figure organization and captions are uninformative and incomprehensible. For example, in figure 2, the differences between the "before" and "after" gray maps seem insignificant. In figure 6, we are shown basically the same map 3 times, with no annotation that might help understand what part we are looking ate and why.

**AC** Thanks for your valuable comments. We have provided a more accessible description for the seismic data processing (line 75-107). We add a new Figure 2 to distinguish differences between the "before" and "after" gray maps (Figure R4 in this response letter) (line 114-119) and detailed descriptions about the differences (line 99-103). We have redrawn the image (Figure 5 in reversed version, Figure R5 in this response letter) and removed duplicate images, highlighting HIWs, bore-like waves and background waves, etc (line 212).

[Figure]

Figure R4 Example from Line 25 showing the effects of the band-pass filtering and notch filtering. (a) Original seismic data for 68-81 km section; (b) After the band-pass filtering and notch filtering; (c) The difference between the (a) and (b); (d), (e) Tracked reflections (blue solid lines) from the (a) and (b), respectively. (f) Slope spectra. The gray (black) line calculated from the tracked reflectors before(after) filtering of the seismic data.

[Figure]

Figure R5 Seismic image of the internal waves. (a) Seismic image of line25. Background waves and mode-2 waves are denoted; (b) The 1.25-33 km section of the image (a). High-frequency waves, bore-like waves and breaking waves are denoted.

**RC1-9** The manuscript is poorly constructed. The authors do not seem interested in explaining meaning and relevance of their data and observations. The "features" they discuss are not corroborated in any way, the reliability of their conclusions is not evaluated. Perhaps the authors are correct but we have no way to tell - only their word for this. For someone that is unfamiliar with seismic maps this is hard to swallow.

**AC** Thanks for your valuable comments. We have made extensive revisions to the manuscript, adjusting its content and structure. The corresponding updates are highlighted in red in the revised version. To increase credibility, we added other datasets including remote sensing data and tidal models (as mentioned above by RC4). We have provided a more accessible description and summarized the main content in the Abstract (line 8-18): The shoaling and breaking of internal waves are critical processes in the ocean's energy cascade and mixing. Using the seismic data, we observed high-frequency internal waves, which were primarily distributed in the depth range of 79-184 m. Their amplitude scale is $O$ (10 m), with half-height widths ranging from 154 to 240 m. The shoaling thermocline and gentle slope with a low internal Iribarren number suggest that high-frequency internal waves observed are likely a result of fission. The remote sensing data supports the point. Instability estimations showed that due to the strong vertical shear, $Ri$ in the range of 20-

30 km was less than 1/4, and KH billows can be found in the seismic transect, suggesting that these waves were unstable and might dissipate rapidly. We use seismic data to estimate diapycnal mixing, and we found that the high-frequency internal waves can enhance diapycnal mixing, averaging $10^{-4}$ $m^2s^{-1}$. The maximum mixing value is up to $10^{-3}$ $m^2s^{-1}$, and it is associated with the breaking of IWs caused by the strong shear. The results show a new energy cascade route from shoaling ISWs to turbulence, i.e., the fission of ISWs into HIWs, which improves our knowledge of ISW energy dissipation and their roles in improved mixing in the northern SCS.

**Responses to Reviewer 2**

The authors (AC) thank Reviewer 2 (RC2) for the valuable comments and good suggestions that greatly helped to improve our manuscript. Below are our itemized.

**RC2 Summary:** This paper presents a section of seismic reflection data from the South China Sea in which the detail of reflection variability is assessed. The presence of high-frequency and high mode internal waves, as well as KH billows are identified, and related to computed turbulent dissipation rates across the section.

**AC** Thank you for your valuable comments. We have carefully considered your comments and have made major revisions to the manuscript to address your concerns. Specifically, we have adjusted the content and structure of the manuscript to enhance its clarity and rigor. The corresponding updates are highlighted in red in the revised version. Below are our itemized.

**RC2-1** There are some interesting features in the seismic data which are worth publication and this is one of the few techniques of mapping out the horizonal detail of such structures. The interpretation of the different features associated with internal wave processes is interesting, however I feel that some parts need to be presented as a little more speculative – particularly the high-mode internal solitary waves. In an ideal world, an acoustic response to a model of such features would be really needed to understand exactly what is shown in the data. To be of an 'international standard' this needs addressing.

**AC** Thank you for your valuable comments and suggestions. Firstly, to increase credibility, we added other datasets including remote sensing data and tidal models to Section 3.2 in the revised manuscript, using red colored texts (line 215-241). MODIS image (Figure 6, Figure R2 in the response letter) revealed that ISWs from the Luzon Strait typically appeared as wave packets near Line 25 during the data collection period, with tidal models indicating these waves originated during neap tide. In addition, previous research shows that fission generally applies to gentle slopes with low $\xi_{in}$ (Aghsaee et al., 2010). Calculations show that the seabed slope corresponding to Line 25 is S = 0.018 (≤0.05), with $\xi_{in}$ ranging from 0.03 to 0.17. Therefore, we infer that HIWs are products of the fission of shoaling ISWs.

Secondly, we use the seismic data to estimate the vertical shear of horizontal velocity and Richardson number. Instability estimations showed that due to the strong vertical shear, $Ri$ in the range of 20-30 km, where KH billows can be found, was less than 1/4, suggesting that these waves were unstable and might dissipate rapidly. We add a new Figure 9 (Figure R6 in the response letter) and detailed descriptions in in the revised version (line 133-149, line 295-327).

Finally, there are two nonlinear waves with complex structures, and their shapes like bores (Figure 13 in the revised version). Given the complexity of the shape, we are not sure what they are

exactly. Therefore, we place this part in the discussion section (Section 4.1, 4.2, line 376-413).

[Figure]

Figure R6 Geostrophic shear and Richardson number (Ri) estimated from seismic data. (a) Geostrophic shear (in color) are superimposed on the seismic images (in gray). (b) represents the mean vertical shear (blue line) and standard deviation (blue shading) for the depth of 50-200 m of the seismic transect. (c) Richardson number (Ri) are superimposed on the seismic images (in gray).

**RC2-2** A little more input around why the identification of these kind of features may be important (i.e. in the wider context) would be good, alongside what particularly the role of seismic data brings. This is needed to hit putting the 'results into context'.

**AC** Thank you for your valuable suggestions. We have revised the introduction section and restated the research question and significance (why the identification of these kind of features may be important). please see lines 30-57 of the revised manuscript: When the energy transfer from large and mesoscale motions to small-scale turbulence, internal waves (IWs) play a crucial role (Liu et al., 2022) … This represents our most common understanding of the dissipation mechanisms of ISWs … the energy dissipation mechanisms of IWs (ISWs) are complex and the ocean is a multi-scale coupled system with various dynamic processes. However, most current studies consider the ideal scenario where only IWs are present in the ocean's dynamic system, with few considering or only considering simplified dynamic environmental impacts. This results in an incomplete understanding of the shoaling processes and ISWs' dissipation mechanisms. Additionally, whether HIWs, another product of fission, significantly enhance energy dissipation and mixing remains a critical question for comprehending the formation and evolution of wave packets and their role in ocean mixing and energy cascades.

**RC2-3** The paper is quite repetitive in places, especially around the internal wave dynamics in this environment and could be streamlined a little bit.

**AC** Thank you for your valuable suggestions. We have streamlined the repetitive sections, particularly those discussing internal wave dynamics in this environment. The revised manuscript now offers a more concise and focused presentation of the key findings.

**RC2-4** Include how often was velocity picking conducted?

**AC** Thank you for your valuable comments. The velocity picking was comducted every 6.25 km (6.25 m ×1000 CMP).

**RC2-5** 'Further denoising' do you mean trace smoothing as this can impact displacement spectra.

**AC** Thank you for your valuable comments. 'Further denoising' can impact the displacement spectrum to some extent, but the primary goal of this step is to suppress ambient noise and harmonic noise, ensuring the seismic frequency band carries the best possible turbulence information. We add new Figure 2 (Figure R4 in this response letter) and more detailed descriptions in revised manuscript (line 75-103).

**RC2-6** Line 47 – around here it would be good to include a couple of sentences explaining what seismic oceanography is (how it works), exactly what it captures, and its resolution. i.e. move lines 79 to 83 in here.

**AC** Thank you for your valuable suggestions. We add more detailed descriptions of seismic oceanography in revised manuscript, using red colored texts (line 58-69).

**RC2-7** Lines 83-85: Have you looked at any nearby hydrographic data to verify the relative temp: salinity contribution, and if seismic reflectors likely track isopycnals?

**AC** Thank you for your valuable comments. The relative temp: salinity contribution has been proved by many authors. Holbrook et al. (2003), Nandi et al. (2004) and Nakamura et al. (2006) have shown that the reflectors imaged correspond indeed to oceanic thermal structures. and, Ruddick et al. (2009) have shown that temperature variations have the dominant contribution to acoustic impedance contrasts and that salinity variations strengthen impedance contrasts by O (10%). Sallarès et al. (2009) determine that the mean contribution of density variations is 5–10%, while 90–95% is due to sound speed variations. On average, 80% of reflectivity comes from temperature contrasts. Salinity contribution averages 20%, but it is highly variable and reaches up to 40% in regions prone to diffusive convection. Zhang et al. (2022) shows that mean contribution to the reflection coefficient comes from sound speed and temperature variations, which account for 81% ($R_V/R$) and 71% ($R_T/R$), respectively. Therefore, as a low-latitude region, except for the double diffusion region , the mean contribution to the reflection coefficient comes from sound speed and temperature variations in the South China Sea.

Krahmann et al. (2009) demonstrated that the comparison of the temporal/spatial slopes of CTD-derived reflectors with those of isopycnals shows a good agreement when the slopes are determined over intervals shorter than 4 hours. Therefore, in this paper, given that the average speed of the vessel is 2.5 m/s, within a certain length range (~36 km), these reflections coincides with the isopycnal. During the calculations for geostrophic shear and mixing, the window' length is less than 36 km.

**RC2-7** Data and methods – include a paragraph on the hydrographic data used i.e. when collected.

**AC** Thank you for your valuable suggestions. We have added a paragraph on the hydrographic data, using red colored texts (line 120-131).

**RC2-8** Line 164: this range will vary depending on dissipation rates – is it ok to just use a set upper and lower wavenumber bound?

**AC** Thank you for your valuable comments. We partially agree with your viewpoint. Theoretically, the wavenumber range for the turbulence subrange of the displacement spectrum may indeed vary across different regions. However, these spectra exhibit statistical regularities, with their average spectrum aligning well with the Batchelor model spectrum. Therefore, we select a unified turbulent wavenumber range based on the average spectrum, which represents a statistically significant wavenumber interval. References such as Holbrook et al. (2013), Gong et al. (2021), and Tang et al. (2020, 2021) have also used this method to calculate turbulent diffusivity.

**RC2-9** Fig 4- there seems to be some odd jumps in the black lines between reflectors e.g. distance 25 km and distance ~ 46 km, depth ~400 m.

**AC** Thank you for your valuable comments. The black lines represent the picking reflectors. We have added a new picture (Figure 4a in revised manuscript, Figure R7 in this response letter) in line 185-191. As is shown in Figure 13 (Figure R8 in this response letter), we observe the nonlinear waves (NIWs) with complex structures, and their shapes like bores at the distance 25 km, depth ~400 m. We will study this complex structure in the discussion section (line 376-413). At the distance ~46 km, depth ~400 m, the reflectors show large inclinations or even break as a result of the interaction between waves and seabed.

[Figure]

Figure R7 Picked reflections and the slope spectrum. (a) Picked reflections (blue solid lines) from the seismic Line 25

[Figure]

Figure R8 Bore-like NIWs. (a) Enlarged view of the 19-27 km transects in Fig. 5; (b) Interpretative diagram of the reflectors in the seismic image, where red solid lines represent the boundaries of NIWs, black solid lines represent HIWs, and blue dashed lines indicate hydraulic jump.

**RC2-10** Fig4 – what are the slopes of the fitted red lines? A pdf would be good to check they align with Batchelor. I don't understand what the lower wavenumber, negative slope is fitted to – it does not appear to match the spectra at all?
**AC** Thank you for your valuable comments. We apologize for that this figure is unclear. As is shown in Figure R9 (Figure 4b in revised manuscript), we have added a new picture to interpret the slopes of the fitted red lines. Figure R9 shows the slope spectrum calculated from the reflectors in Fig. R8, with 95% confidence interval. The black dashed lines with -1/2 and 1/3 slopes correspond to the theoretical slopes of GM75 for internal wave subrange and Batchelor model for turbulence subrange, respectively. The black dashed line with -2 slope is the noise subrange. The blue shaded area represents the turbulent subrange for fitting the spectrum.

[Figure]

Figure R9 (b) Slope spectrum calculated from the reflectors in Fig. R8, with 95% confidence interval. The black dashed lines with -1/2 and 1/3 slopes correspond to the theoretical slopes of GM75 for internal wave subrange and Batchelor model for turbulence subrange, respectively. The black dashed line with -2 slope is the noise subrange. The blue shaded area represents the turbulent subrange for fitting the spectrum.

**RC2-11** Fig 2 – it is difficult to assess differences in the stacks – maybe you could show some zoomed regions and highlight some key areas of improvement? Also, what do you mean by before and after processing? i.e. data in a has been processed to a degree?
**AC** Thank you for your valuable comments. We add a new Figure 2 to distinguish differences between the "before" and "after" gray maps (Figure R4 in this response letter) (line 114-119) and detailed descriptions about the differences (line 99-103).

**RC2-12** Fig 7: thermocline in XBT 7 looks deeper than XBT5 and 6? Line 204 – needs more clarification about how you identify the thermocline and what you mean by a significant change?
**AC** Thank you for your valuable comments. We determine the thermocline depth by taking the mid-depth between two inflection points on the temperature profile, where temperature changes significantly. We have added new Figure 8 (Figure R10 in this response letter) show the variations of thermocline depth and the temperature difference with water depth. We can observe that, except

for XBT-7, the thermocline shoals with the shoaling topography, with the depth decreasing from 127 m to 57 m, and the temperature difference between the upper and lower layers decreasing from 8.2°C to 1.8°C. Therefore, the overall trend shows that the thermocline shoals with the shoaling topography.

[Figure]

Figure R10 Variations of (a) thermocline depth and (b) Temperature difference between the upper layer and lower layer with the shoaling water depth

**RC2-1**3 Line 244 – ok but a little bit of an assumption. Could also be lenses? Especially considering your comment in line 260. I think you need to clearer that this is one of other possible interpretations. Noted you come back to this discussion in 4.1 – but I think some kind of modelling is needed to really interpret these structures. I think you need to be more tentative in the identification here e.g. line 395.

**AC** Thank you for your valuable comments. As shown in Figure 13 (Figure R8 in this response letter), we observe the nonlinear waves (NIWs) with complex structures, and their shapes like bores. Given the complexity of these waves, we are not sure what they are exactly. Therefore, we place this part in the discussion section (Section 4.1, 4.2, line 376-413). We discuss this structure in two parts. Firstly, we describe the internal structure of the bore-like nonlinear waves in Section 4.1 (line 376-392): As shown in Figure 13, we observe the nonlinear waves (NIWs) with complex structures…When around 20 km, the reflectors no longer remain horizontal. the reflectors of upper boundary gradually become convex upwards with about a 70 m displacement… Unlike the former, the reflectors at the rear (~ 26 km) do not return to the original stratification but become disordered and breaking, representing the enhanced turbulence state in the ocean.

There are two speculations about what they are exactly, that is, the mode-2 ISWs or shoaling bores. We have added a more detailed section for discussing the complex structures, with red colored texts (Section 4.2, line 393-413): There are two speculations about what they are exactly. Firstly, shoaling internal waves usually form bores with bottom-enclosed structures carrying cold water masses that continue to propagate shoreward after breaking… Bores typically begin to dominate the structure of internal waves near depths less than 50 m, becoming a primary feature of the wave during the shoaling process… According to simulations by Brandt and Shipley. (2014), we think that they might belong to exceptionally large-amplitude internal solitary waves.

**RC2-14** Line 275-280. I'm finding it hard to distinguish between symmetric and asymmetric –

appears very qualitative. Also – have you investigated how migration approaches impact this analysis?

**AC** Thank you for your valuable comments. We regret any confusion caused by our previous explanations. We wish to convey that when less than 28 km, the reflectors are shorter and incomplete, which may be caused by high shear, while the braid structure is more evident and the reflectors are longer at greater than 28 km, as is shown in Figure 10 (Figure R11 in this response letter). The more detailed description has been added to the the revised manuscript in lines 315–319.

The Stolt migration method quickly reveals the true shape and position of reflectors, facilitating subsequent reflectors' picking.

[Figure]

Figure R11 Structure of KH billows. (a) Enlarged view of the transect containing KH billows; (b) Interpretative diagram of the reflectors in the seismic image, where blue solid lines represent KH billows, and red solid lines represent the rear wing of bore-like waves, whose waveform is not very complete due to instability.

**RC2-15** Fig 12 – does this account for all the errors noted or just the spectral fitting ones?
**AC** Thank you for your valuable comments. We move the error analysis section to Appendix A, and Fig 12 becomes Fig A1 (line 447-458). It only account for the spectral fitting error.

**RC2-16** More generally – does the internal wave part of the spectra tell you anything about the character of these waves to support your conclusions?
**AC** Thank you for your valuable comments. As is shown in Figure R12 (in this response letter), The energy of the internal wave spectrum in this subrange is higher than that of the GM spectrum, indicating that the internal wave energy in this region is relatively strong. This result supports our view, indicating that internal waves are well-developed in this region.

[Figure]

Figure R12 Slope spectrum calculated from the reflectors in Fig. 4a, with 95% confidence interval. The black dashed line with 1/3 slopes corresponds to and Batchelor model for turbulence subrange, respectively. The green dashed line represent the GM75 towed spectrum for internal wave subrange. The black dashed line with -2 slope is the noise subrange. The blue shaded area represents the turbulent subrange for fitting the sprectrum.

**RC2-17** Line 403 – the increased mixing in fig 10 does need seem to strongly coincide with the region of KH billows identified? A pdf of mixing rates for billow vs non-billow region would be good to see.

**AC** Thank you for your valuable comments. As is shown in Figure 11, a series of shoaling events occurred within the 1.25-38.75 km range: the HIWs can enhance the diapycnal mixing, averaging $10^{-4}$ m²s⁻¹; In the range of 20-30 km, a marked increase in $K_\rho$ suggests that enhanced mixing associated with breaking of IWs caused by the strong shear (Figure R6 in this response letter). The red patches have a slightly deeper color than the other area, indicating that this is a high mixing area in the transect, up to $10^{-3}$ m²s⁻¹. And when away from the topography (62.5-88.75 km), the mixing weakens. The histogram of the diapycnal mixing in Figure 12 suggests the mixing for the 1.25-38.75 km range is $10^{-3.84\pm0.11}\ m^2 s^{-1}$, which is 3.5 times greater than the calculation for the 62.5-88.75 km range ($10^{-4.39}\ m^2 s^{-1}$).

The strong shear in Figure 9 (Figure R6 in this response letter) and the braids structures in Figure 10 (Figure R11 in this response letter) show the KH billows are distributed in the range of 26.5-30 km at the depth of 100-200 km. The region of KH billows identified coincide with elevated mixing. Figure R13 is the pdf of mixing rates for billow vs non-billow region.

[Figure]

Figure R13 (a) the pdf of mixing rates for billow region (distance: 26.5-30 km, range: 100-200 m); (b) the pdf of mixing rates for non-billow region (distance:80-88 km, depth: 200-400 m).

**RC2-18** Remove 'And' in line 16.

**AC** Thank you for your valuable comments. We have removed 'And' in line 16.

**RC2-19** Fig 10 – check this colour bar is ok with colour blindness.

**AC** Thank you for your valuable comments. We used the classic jet colormap, which is a widely used color mapping algorithm in computer vision. It provides high contrast, effectively highlighting details in the image. A lot of articles have used this color scheme (e.g., Dickinson et al., 2017; Yang et al., 2023).

**Responses to Reviewer 3**

The authors (AC) thank Reviewer 3 (RC3) for the valuable comments and good suggestions that greatly helped to improve our manuscript. Below are our itemized.

**RC3 Summary:** This paper is an attempt to use seismic reflection data over the continental slope (note I would call this region the slope, not the shelf) in the SCS to calculate the horizontal wavenumber and then the TKE and diapycnal diffusivity for the "high frequency" thermal displacements observed. The technique is sound, as has been previously documented in Holbrook (2003, 2013) and the nice how-to by Ruddick et al., 2009 in Oceanography (22). It's an attractive way, perhaps the only way, of estimating these quantities efficiently over a large spatial area. The numbers are probably solid and the distributions interesting. The scientific interpretation of the fluctuations observed in the seismic data needs to be improved a lot to be publishable in NPG.

**AC** Thank you for your valuable comments. We have carefully considered your comments and have made major revisions to the manuscript to address your concerns. Specifically, we have adjusted the content and structure of the manuscript to enhance its clarity and rigor. The scientific interpretation of the fluctuations observed have been improved. The corresponding updates are highlighted in red in the revised version. Below are our itemized.

**RC3-1** The largest, can't-miss-it oceanographic signal by far in this part of the world is the mode-1 NLIWs with amplitudes from 100-150m, as described by many previous authors. Why are they not observed in this data set? Note there may be explanations for this, for instance observations made during neap tide in the Luzon Strrait, but the authors need to address this.

**AC** Thank you for your valuable comments. In fact, we have observed large amplitude ISWs on other seismic lines within the same dataset, and have published several articles on these findings. As shown in Figure R14 (in this response letter), Geng et al. (2019) observed large amplitude ISWs on other lines from the same cruise. These ISWs are primarily distributed across the continental slope at depths ranging from 263 to 740 m, with maximum amplitudes between 35 and 128 m. The results indicate a significant correlation between the maximum amplitude of ISWs on the northern slope of the South China Sea and the seabed depth, presenting a logarithmic relationship. As illustrated in Figure R15 (in this response letter), Gong et al. (2021a) described and analyzed 11 ISWs from the same cruise, comparing their vertical structures with two theoretical models (the linear vertical structure function and the first-order nonlinear vertical structure function). The results demonstrated that the vertical structure is primarily determined by nonlinearity, which is a crucial dynamic characteristic of ISWs. Using data from the same cruise, Kuang et al. (2021) conducted a study combining seismic and remote sensing data on an ISW wave packet. This study included the

analysis of waveform characteristics and kinematic properties such as vertical structure, phase speed, and half-height width.

In summary, these symmetrical, large-amplitude ISWs are predominantly found in deeper waters at depths ranging from 350 to 700 m, distant from the seabed topography. In contrast, our manuscript focuses on the high-frequency internal waves (HIWs) and their energy dissipation mechanisms over shoaling terrains (<300 m). We have updated the figure to marked the locations of the lines studied (Figure 1 in the revised version). Additionally, remote sensing data and tidal models (Figure R2 in this response letter) indicate that these HIWs likely originate from solitons generated in the Luzon Strait during neap tide periods.

[Figure]

Figure R14 (a) Stacked section, (b) common offset section, (c) picked shallow ISW reflections after NMO corrections and ISW's amplitude definition, and (d) vertical amplitude distribution versus water depths of L1. The red lines indicate the picks of the ISW reflection events. The amplitudes are combined from stacked and common offset sections. (From Figure 2 in Geng et al., 2019).

[Figure]

Figure R15 The seismic sections (left panel) and vertical structures (right panel) of ISW1-1 (a), ISW4-3 (b), ISW6 (c) and ISW7 (d). (From Figure R13 in Gong et al., 2021a).

**RC3-2** Nobody has ever seen mode-2 NLIW this large, especially in the absence of mode-1 waves, so I have a hard time believing this interpretation. Are these "mode-2" waves perhaps misinterpreted mode-1 waves?

**AC** Thank you for your valuable comments. Indeed, there have been no previous reports of such large second-mode nonlinear waves. The structure we observed using seismic data is notably complex. On one hand, their shapes resemble that of a shoaling bore. Although its depth distribution does not align with typical bore depths (<50 m), we cannot entirely rule out this possibility without further evidence. On the other hand, the reflective structure exhibits an upward convex upper boundary and a downward concave lower boundary, indicative of a high-mode structure. As shown in Figure R16, Brandt and Shipley's (2014) simulation results suggest that this could be a second-mode wave with exceptionally large amplitude.

In Figure R17, R18, Scotti et al. (2008) found that strong nonlinear high-frequency internal waves (HIWs) interacting with shoaling topography in Massachusetts Bay can lead to "Bottom Collision Events (BCEs)." Specifically, they observed that under conditions of gentle slopes and moderate amplitude, these waves undergo high-energy collision events, producing waveforms similar to those observed in Figure 13 (Figure R8 in this response letter). Thus, we propose that this phenomenon observed in the seismic data could result from the strong interaction between internal waves and topography.

Finally, Given the complexity of the structure, we have included the observations and analysis of this section in the discussion part of the manuscript. The updated text can be found in the revised manuscript on lines 376-413.

[Figure]

Figure R16 The simulation results' by Brandt and Shipley's (2014). Black and white images of representative ISW bulges, (a) lower amplitude; (b) large amplitude; (c) very-large amplitude.

[Figure]

Figure R17 The simulation results' by Scotti et al. (2008). The approaching nonlinear waves interact strongly with a shoaling bottom. (a) Density field at 7 h; (b) Density field at 9.5 h; (c) Density field at 11 h. The isolines contoured correspond to 22.5, 23.5, and 24.5. Horizontal distances are in kilometers and depth in meters.

[Figure]

Figure R18 Examples of NLIWs observed in the temperature record (After Scotti et al. (2008)). An increase in temperature denotes downward displacement of isotherms. Figure R16a shows the passage of an undular bore, similar in shape and properties to the ones that are normally observed propagating in the deeper section of the Bay. Figures R16b–R16d show examples of BCEs, caused by the approaching undular bore undergoing shoaling.

**RC3-3** Small amplitude (10-15m) random internal waves are everywhere in the SCS. They did not necessarily form by shoaling/breaking. In fact, all the literature dedicated to shoaling (Sinnett et al. 2022 and similar) are talking about the very large mode-1 NLIWs which occupy a significant fraction of the water column. The fission waves are always seen trailing the mode-1 soliton. There is no reason to believe the small linear IWs described here feel the bottom at all at these depths. I think the internal waves referred to here are just a snapshot of the random IW field in the SCS.

**AC** Thank you for your valuable comments. As stated in our response to **RC3-1**, we discovered large-amplitude ISWs along other lines of the same cruise. These ISWs are primarily distributed in deep water areas with depths ranging from 350 to 700 m, with amplitudes between 35 and 120 m. This paper focuses on atypical internal waves (high-frequency internal waves, or HIWs) and their role in energy dissipation.

Firstly, to differentiate HIWs from random internal waves in seawater, we updated the seismic images (Figure 5, Figure R5 in this response letter). As shown in Figure 5a (Figure R5a in this response letter), background internal waves predominantly develop in the horizontal range of 70-80 km at depths above 400 m. In contrast, HIWs are mainly distributed on the left side of the seismic image (<20 km). Unlike random internal waves, the internal waves developing on the left side of the image exhibit high-frequency oscillations. Secondly, in section 3.2 (line 215-241), we added MODIS images (Figure R2 in this response letter), revealing that during the data acquisition period, ISWs packets propagated northwest across Line 25 on the shoaling shelf. The HIWs observed in the seismic image might have resulted from the fission of these waves. However, due to the limited resolution (250 meters), these features cannot be fully identified from the remote sensing images. Additionally, the shoaling thermocline and low internal Iribarner number on the gentle slope indicate that the observed HIWs could be a result of this fission. Bai et al. (2013, 2019) found that ISWs on the South China Sea shelf also undergo fission during shoaling, generating HIWs, and proposed a new hypothesis for energy dissipation. While they did not directly observe strong-amplitude first-mode internal waves, their numerical simulations supported this viewpoint.

We estimated the mixing parameters of seawater using seismic data and found that these HIWs cause strong turbulent dissipation and energy mixing. The results suggest a new energy cascade pathway from shallow-water ISWs to turbulence, namely the fission of ISWs into HIWs. This enhances our understanding of ISW energy dissipation and its role in improving mixing in the northern South China Sea.

**RC3-4** Upslope bores are limited to very shallow (less than 50 m) water. I wouldn't call anything I can see here a "bore."

**AC** Thank you for your valuable comments. This phenomenon is quite interesting. Although their shapes resemble that of a bore, as you mentioned, bores typically occur in shallow waters with depths less than 50 m. However, this particular structure has indeed been observed by seismic data. Given its complexity and our current limitations in studying it comprehensively, we have decided to place this part of the analysis in the discussion section (Section 4.1,4.2, line 376-413).

**RC3-5** Likewise, I can't see anything in this data I would convincingly call a KH billow. The authors need to somehow highlight these better, and more convincingly.

**AC** Thank you for your valuable comments. To enhance credibility, we calculated the vertical shear of horizontal velocity and the Richardson number using seismic data; the specific method is detailed

in lines 133-149. As shown in Figure 9a (Figure R6a in this response letter), within the horizontal range of 20-30 km on the seismic transect where bore-like waves and billows develop, the reflectors exhibit significant tilting and even fracturing, resulting in maximum shear forces of up to 0.03 s⁻¹. In Figure 9b (Figure R6a in this response letter), regions with Ri < 0.25 comprise approximately 8% of the seismic image. Specifically, within the 20-30 km range at depths of 150-250 m, weak stratification and strong shear result in Ri values less than 0.25, indicating shear instability in the ocean. This coincides with the observed location of Kelvin-Helmholtz (KH) billows at 26.5-30 km. We have updated text in revised manuscript, with red colored texts (line 296-326).

**RC3-6** Line 16 – delete "and"
**AC** Thank you for your valuable comments. We have deleted the "and".

**RC3-7** Line 90 figure 2:  I really can't see any difference between a) and c).
**AC** Thank you for your valuable comments. As stated in **R1-8**'s reply, we add a new Figure 2 to distinguish differences between the "before" and "after" gray maps (Figure R4 in this response letter) (line 114-119) and detailed descriptions about the differences (line 99-103).

**RC3-8** Line 115 - Seems like M-H Chang more recent papers near Dongsha should be referenced.
**AC** Thank you for your valuable suggestions. We cited articles from Professor. Chang's laboratory on KH billows, such as Acabado et al. (2021) and Chen et al. (2022).
* * *
**RC3-9** Figure 6.  Aren't b) and c) the same as a)?  Suggest drawing boxes around the region of interest.  I am hard pressed to see any mode-2 waves or KH billows here.
**AC** Thank you for your valuable suggestions. In the revised version, the original Figure 6 has been replaced by new Figure 5 (Figure R5 in this response letter) in the revised manuscript (line 214). Figure 5a (Figure R5a in this response letter) presents the seismic profile of Line 25, while Figure 5b (Figure R5b in this response letter) shows an enlarged section of the image spanning 1.25-33 km. High-frequency internal waves, bore-like waves, and breaking waves are highlighted. To further illustrate mode-2 waves or Kelvin-Helmholtz (KH) billows, we enlarged the sections from 20-26 km and 26.5-30 km and provided seismic structure interpretation diagrams, as shown in Figures 10 (Figure R8 in this response letter) and 13 (Figure R11 in this response letter), respectively.

**RC3-10** Line 207:  These IWs are quite small, compared to previous literature.  Where are the mode-1 solitons?  These should be the most obvious signal.
**AC** Thank you for your valuable comments. As stated in our response to **RC3-1**, we have discovered large-amplitude ISWs along other lines of the same cruise (Figure R14, R15 in this response letter). These ISWs are primarily distributed in deep water areas with depths ranging from 350 to 700 m, with amplitudes between 35 and 120 m. This paper focuses on atypical internal waves (high-frequency internal waves, or HIWs) and their role in energy dissipation. For further studies on the high-amplitude internal waves observed during this cruise, please refer to Geng et al. (2019) and Gong et al. (2021a). Our research builds upon their findings. We have supplemented our revised manuscript with summaries of their work. The updated text can be found in lines 266-272.

**RC3-11** Lines 212-214:  Three lines of text on "dunes."  There are entire papers on this

phenomenon. The authors need to strengthen this argument considerably or just leave it out. What's the data source of the bottom profile?   Why does it matter for this paper?

**AC** Thank you for your valuable comments. The original intention was to explain that the high-frequency internal waves generated by shoaling can cause sediment resuspension due to strong current shear, thereby reshaping the seafloor topography. This explanation is supported by the studies of Geng et al. (2024). Although seismic data can also image the seafloor, we chose to cut the seafloor data while retaining the information about dunes. However, considering the current feedback, we have decided to remove this section for clarity.

**RC3-12** Line 230:   There are many other references on the impact of the bottom slope on wave evolution.

**AC** Thank you for your valuable suggestions. We cited other references on the impact of the bottom slope on wave evolution (line 281).

**RC3-13** Line 258:   Nobody has ever seen mode-2 NLIW this large.   Are these perhaps misinterpreted mode-1 waves?

**AC** Thank you for your valuable comments. As stated in **R3-2**'s reply, there have been no previous reports of such large second-mode nonlinear waves. The structure we observed using seismic data is notably complex. On one hand, the reflective structure exhibits an upward convex upper boundary and a downward concave lower boundary, indicative of a high-mode structure. On one hand, their shapes resemble that of a shoaling bore.

   This phenomenon is interesting and has not been reported in previous studies. Given the complexity of the structure, we have included the observations and analysis of this section in the discussion part of the manuscript. The updated text can be found in the revised manuscript on lines 376-413.

**RC3-14** Figure 9.   I can't see the difference between symmetric vs. asymmetric billows. We need more description of what is meant by this.

**AC** Thank you for your valuable comments. We regret any confusion caused by our previous explanations. As stated in **R2-14**'s reply, we wish to convey that when less than 28 km, the reflectors are shorter and incomplete, which may be caused by high shear, while the braid structure is more evident and the reflectors are longer at greater than 28 km, as is shown in Figure 10 (Figure R11 in this response letter). The more detailed description has been added to the revised manuscript in lines 315–319.

**Responses to Reviewer 4**

The authors (AC) thank Reviewer 4 (RC4) for the valuable comments and good suggestions that greatly helped to improve our manuscript. Below are our itemized.

**RC4 Summary:** This paper aims at identifying the structure of nonlinear motions (internal solitary waves) responsible for mixing on the shelf and slope of the South China Sea using marine seismic observations. As clearly stated in the introduction of the paper, the structure and formation mechanisms of these motions have been already identified in numerical simulations and the

challenge here is to perform a similar analysis from field measurements using seismic data. To my opinion this paper deserves to be published after these comments, and the other ones below, have been taken into account in a revised version.

**AC** Thank you for your valuable comments and suggestions. We have carefully considered your suggestions and have made significant revisions to the manuscript accordingly. The corresponding updates are highlighted in red in the revised version. Below are our itemized.

**RC4-1** The conclusion in this respect is poorly written and effort should be made to summarize the novelty of the results with respect to the literature.

**AC** Thank you for your valuable comments and suggestions. We have rewritten the conclusion to clearly summarize the novelty of our results compared to previous studies. The corresponding revisions can be found in the revised manuscript, highlighted in red, from lines 427 to 446.

**RC4-2** The paper also needs to be tightened.

**AC** Thank you for your valuable suggestions. We have reviewed the entire manuscript and removed redundant and repetitive content to improve readability and conciseness, particularly those discussing internal wave dynamics in this environment. Please refer to the revised manuscript for detailed changes.

**RC4-3** At many instances indeed, adjectives such as 'strong' (e.g. l. 25), 'relatively small' (l. 191), 'large and medium-scale' (l. 365), etc. are used. These terms should be made quantitative or the authors should say strong/small etc. with respect to what.

**AC** Thank you for your valuable comments and suggestions. We have revised the manuscript to provide quantitative descriptions where possible. Line 25 (line 23 in the revised version): We delete the "strong" word. Line191 (line 312-315 in the revised version): We adjusted the sentence structure and changed it to a clearer statement. Line365 (line30 in the revised version): The "large and medium-scale" has been replaced by "large and mesoscale motions". Large-scale motions in the ocean typically span over 1000 km and persist for several months to years, involving phenomena like ocean gyres and equatorial currents with velocities of 1-10 cm/s. In contrast, mesoscale motions cover tens to hundreds of kilometers and last for several days to months, encompassing ocean eddies and fronts with velocities of 10-100 cm/s. Both scales significantly influence oceanic physical processes and ecosystems.

**RC4-4** l. 28: explain what "shoaling" means.

**AC** Thank you for your valuable comments. As ISWs propagate from deep to shallow waters, they interact with the seabed topography, a process known as shoaling (Sinnett et al., 2022). We have added the interpretation in the revised manuscript (line 32-33).

**RC4-5** l. 34: what do you mean by "critical depth"?

**AC** Thank you for your valuable comments. ISWs in deep water are typically observed as depression waves, while in shallow water, they appear as elevation waves. In the idealized two-layer fluid model, the region where the thickness of the upper layer is less than that of the lower layer is defined as deep water, whereas the opposite is considered shallow water. The boundary between these two regions is referred to as the critical depth or the polarity critical point. Therefore, at critical

depths, the rear of a depression wave steepens and undergoes polarity reversal to form an elevation wave. Please refer to Bourgault et al. (2007), Fu et al. (2012), Liu et al. (1998) and Orr and Mignerey. (2003).

**RC4-6** l. 35: "reversing polarity to form elevated waves": this is not clear, please explain better (or refer to a sketch in a published paper)

**AC** Thank you for your valuable comments. "Reversing polarity to form elevated waves" refers to a phenomenon where the characteristics of an ISW change as it propagates at the critical depth. In simpler terms, it involves the transformation of the wave's profile from a depression wave (where the wave trough is below the undisturbed water level) to an elevation wave (where the wave crest is above the undisturbed water level). Polarity reversal can be observed through various methods, including seismic imaging, satellite observations, and direct field measurements (e.g., Shroyer et al., 2009). As is shown in Figure R19 (in this response letter), it developed from a depression wave through transition to an elevation wave. We have added more detailed description of the "reversing polarity to form elevated waves" in revised version (line 34-37).

[Figure]

Figure R19 (a) Acoustic backscatter, (b) horizontal velocity, and (c) vertical velocity shown for four different periods along the propagation path. The wave is propagating to the right (After Shroyer et al., 2009).

**RC4-7** l. 36: "dispersion continues to form elevated waves": hard to understand if "elevated waves" is not explained.

**AC** Thank you for your valuable comments. Elevation waves are a type of internal solitary wave characterized by a crest that is elevated above the undisturbed water level. In other words, these waves cause a peak in the water column, rather than a trough. Figure R20 illustrates the sign of vorticity for (a) depression and (b) elevation waves.

[Figure]

Figure R20 Schematic illustrating the sign of vorticity for (a) depression and (b) elevation waves (After Shroyer et al., 2009).

**RC4-8** l. 45: references are needed after "through numerical simulations".

AC Thank you for your valuable comments. We have add the references about numerical simulations. Such as Liu et al. (1998), Terletska et al., (2020), and Vlasenko et al. (2002).

**RC4-9** l. 50: "we attempt to use" -> change into: we use.

AC Thank you for your valuable comments. We have removed the words "attempt to".

**RC4-10** l. 51: "we were fortunate to observe" -> change into: we were able to observe.

AC Thank you for your valuable comments. We have changed the sentence.

**RC4-11** This is a very technical section and one reference (at least) is requested, for instance at the end of the second paragraph, on l. 65.

AC Thank you for your valuable suggestions. We have added the references Ruddick et al. (2009) and Holbrook et al. (2013) in the Seismic data acquisition and processing of the revised manuscript. (line83-84)

**RC4-12 l. 59:** "red solid line" -> black solid line, actually.

AC Thank you for your valuable comments. We have corrected it. It is noted that we have added the new Figure (Figure 1 in the revised manuscript, Figure R21 in this response letter), and the blue solid line represents the line 25.

**RC4-13** The caption should be understandable by the reader, which is not the case here. Indeed:
• "the line 25" should be explained or a reference to the line of the paper where this is explained should be provided.
• the difference between the "magenta" and the "red" colors is not easy to see -> use another colour for magenta (orange for instance).
• what does mean "calculated to have a seabed slope": is it just the calculation of the slope which yields this value of $\gamma$ ?
• About this parameter $\gamma$: there is a confusion between the angle and the slope (equal to the tangent of the angle) which are both denoted by $\gamma$. Thus "seabed slope $\gamma =1.08°$ (0.018)" should be replaced by "seabed angle 1.08° (associated with the slope $\gamma =0.018$)".

AC Thank you for your valuable comments. We have added a new figure (Figure 1 in the revised manuscript, Figure R21 in this response letter).
1.  "the line 25" is the seismic line 25, we have added the introduction about it (line 75-77).
2.  We have used the "green" color to replace the "magenta" color.
3.  We are very sorry that our statement was incorrect. The slope is 0.018, which was measured during the data collection process of the survey vessel.
4.  In the revised version, we have uniformly used the term "slope", that is, "$\gamma =0.018$".

[Figure]

Figure R21 Bathymetry map of the South China Sea and seabed topography along seismic line 25. (a) Topographic map of the research area. The blue solid line represents the seismic line 25, red dots are XBT data (numbered from right to left, XBT1-XBT9), green dots are CTD data (numbered from right to left as CTD1-CTD4). The black solid lines have been studied by Geng et al., (2019) and Gong et al. (2021a); (b) Seabed topography along Line 25. The slope γ = 0.018.

**RC4-14** Caption of Figure 2: Which quantity is displayed in each panel? Is it the isopycnal displacement? This should be said. Also, avoid writing "F-K" spectrum and write explicitly "frequency-horizontal wavenumber spectrum".

**AC** Thank you for your valuable comments. As stated in **R1-8**'s reply, we add a new Figure 2 to distinguish differences between the "before" and "after" gray maps (Figure R4 in this response letter) (line 115-119) and detailed descriptions about the differences (line 99-103).

**RC4-15** l. 373: what do you mean by "global instability"?

**AC** Thank you for your valuable comments. We correct the statement in the line 419: Instability near the seabed boundary layer causes sediment resuspension and transport

**RC4-16** l. 381 and 384: "diapycnal mixing of the order of 10-4", "the order of 10-3": add physical unit.

**AC** Thank you for your valuable comments. We have added the unit.

**RC4-17** l. 388: "HISW" has not been defined previously.

**AC** Thank you for your valuable comments. We have replaced the word with bore-like nonlinear wave. Given the complexity of the structure, we have included the observations and analysis of this section in the discussion part of the manuscript. The updated text can be found in the revised manuscript on lines 376-413.

**RC4-18** As indicated in the first general comment above, the conclusion is poorly written and should be rephrased following my comment.

**AC** Thank you for your valuable comments. As stated in **R4-1**'s reply, we have rewritten the conclusion to clearly summarize the novelty of our results compared to previous studies. The

corresponding revisions can be found in the revised manuscript, highlighted in red, from lines 427 to 446.

**Responses to Reviewer 5**

The authors (AC) thank Reviewer 5 (RC5) for the valuable comments and good suggestions that greatly helped to improve our manuscript. Below are our itemized.

**RC5 Summary:** This has the potential to be a very interesting study adding to the growing literature of seismic oceanography. However, I have several concerns about the observations and interpretation, and the lack of clarity on the overall contribution:

- Needs more work to verify that these are ISWs from the Luzon Strait either with corroborating data (such as satellite images, tidal information) and/or comparing to other seismic reflection lines to verify what the background noise is and that Line 25 deviates significantly from the background. If it cannot be convincingly shown that these are internal waves, the claims in this study should be treated with far more skepticism.

- Add in more of a discussion of why this study is important. I do think it has the potential to contribute to seismic oceanography, but please emphasize this more and add more context. For example, if more work is done to demonstrate that these are ISWS, emphasize that. I do not agree that this is well-established and that alone would be an important result.

**AC** Thank you for your valuable comments and suggestions. We have carefully considered your suggestions and have made significant revisions to the manuscript accordingly.

Firstly, to verify that these ISWs originated from the Luzon Strait, we have included remote sensing data and tidal models (lines 215-241, Figure R2 in this response letter). The results show that during the data acquisition period, ISW packets passed through line 25 (the studied line) and propagated northwest over the shoaling continental shelf. The tidal model indicates that this wave packet likely originated from neap tides in the Luzon Strait.

Secondly, as mentioned in our response to **RC3-1**, we observed large-amplitude ISWs on other seismic lines during the same cruise (Figures R14, R15 in this response letter) (e.g., Geng et al. (2019), Gong et al. (2021a)). In this paper, we primarily focus on atypical internal waves (high-frequency internal waves) and their energy dissipation. To distinguish between random internal waves (background waves) and high-frequency internal waves, we have redrawn the seismic images (Figure 5, line 212).

Lastly, we have made significant modifications to the structure and content of the paper to emphasize the research objectives and significance. The corresponding updates are highlighted in red in the revised version. Below are our itemized responses.

**RC5-1** My main concern is that the study starts with the assumption that nonlinear internal waves are being detected. However, I am not convinced. I agree that the seismic reflection methodology is well-established, however, the detection of internal waves using this method is not. For example, the authors state that seismic oceanography has been extensively used in the study of internal solitary waves, but only provide one reference to back up this claim. Further, the Tang et al. (2014) reference provided observes two much clearer examples of internal solitary waves of depression that are then corroborated by a satellite image. These waves are convincingly leading waves in a

packet of Mode-1 ISWs. The proposed internal wave detections here are not as convincing and should be corroborated with other data before they are interpreted.

**AC** Thank you for your valuable comments and suggestions.

1. We have supplemented more references about the detection of internal waves using this seismic method (line 66). In fact, due to the unique advantages of seismic methods in studying the vertical structure of water column (as mentioned above in **RC2'** response), this method has been widely used in the study of ISWs, and has achieved some significant research results. The relevant research is as follows: Tang et al. (2014) used pre-stack seismic data to calculate the apparent phase velocity of ISWs on the northeastern slope of the South China Sea, correlating this with hydrological and remote sensing data to analyze wave characteristics. Tang et al. (2015) optimized phase velocity calculations by grid search, selecting shot-receiver pairs with the best slope for the wave's leading edge. They calculated kinematic and dynamic parameters of the wave packets northeast of Dongsha Island and found that these waves align better with the KdV theory. Bai et al. (2015) used seismic methods to study the structural characteristics of ISWs in the Dongsha area, finding their energy in the low-wavenumber range to be about two orders of magnitude higher than the GM model spectrum. Tang et al. (2016) combined hydrological and reflection seismic data with the Markov chain Monte Carlo method to detail the fine structure of temperature and salinity in seawater, achieving detailed imaging of the ISWs. Bai et al. (2017) analyzed the polarity conversion process of shoaling ISWs in the northeastern South China Sea using reflection seismic data. Huang et al. (2018) combined fluid dynamics numerical simulations and reflection seismic forward modeling to study the geometric and structural characteristics of seawater layers during different stages of shoaling. Tang et al. (2018) used high-resolution reflection seismic imaging and simultaneous hydrological data to invert secondary ISWs on the northern shelf of the South China Sea, directly observing KH instabilities. Geng et al. (2019) combined seismic data with KdV theoretical models to study the vertical structure of ISWs, finding that their maximum amplitude decreases logarithmically with depth. Gong et al. (2021a) used linear and nonlinear vertical structure theories to study ISWs in the northeastern South China Sea, concluding that nonlinearity, topography, and background flow influence their vertical structure. Sun et al. (2019), Fan et al. (2020), and Kuang et al. (2021) used marine seismic methods to study the structure and propagation of ISWs near the Strait of Gibraltar and the northeastern South China Sea. Gong et al. (2021b) captured three polarity-reversing ISWs near Dongsha Island using seismic sections, discovering that the mixing rate induced by these waves is two orders of magnitude higher than in the open ocean, with the diapycnal mixing rate of polarity-reversing waves being three times higher than non-reversing regions. Fan et al. (2021, 2022) used reflection seismic data to discover numerous shoaling mode-2 ISWs in Central America, analyzing their amplitude and propagation speed relationship and regional distribution. Song et al. (2021b) observed the shoaling process of ISWs through common-offset gathers pre-stack migration profiles, documenting changes in phase velocity, amplitude, wavelength, and slopes, further enhancing our understanding of ISWs and their shoaling processes. In a word, the seismic method has been widely recognized.

2. We have supplemented remote sensing data and tidal models (lines 215-241, Figure R2 in this response letter). The results show that during the data acquisition period, ISW packets passed through line 25 (the studied line) and propagated northwest over the shoaling continental shelf. The tidal model indicates that this wave packet likely originated from neap tides in the Luzon

Strait. In addition, we have observed large-amplitude ISWs on other seismic lines during the same cruise (Figures R12, R13 in this response letter), please refer to Geng et al. (2019) and Gong et al. (2021a). Our research builds upon their findings. We have supplemented our revised manuscript with summaries of their work. The updated text can be found in lines 266-272.

**RC5-2** I am also confused as to why the proposed high-frequency internal waves are classified as mode-2 waves. Where are the mode-1 waves, which should be the largest signal, and were detected by Tang et al. (2014)? I realize this survey line is a snapshot in time – do the authors propose that the mode-1 wave has already passed? If so, please discuss further.

**AC** Thank you for your valuable comments. It appears there was a misunderstanding regarding the observed high-frequency internal waves. We think that these waves are generated by the fission of shoaling depression waves, rather than being mode-2 waves. In fact, as mentioned in our response to **RC3-1**, we observed large-amplitude ISWs on other seismic lines during the same cruise (Figures R14, R15 in this response letter). These symmetrical, large-amplitude ISWs are predominantly found in deeper waters at depths ranging from 350 to 700 m, with maximum amplitudes between 35 and 128 m, distant from the seabed topography. In contrast, our manuscript focuses on the high-frequency internal waves (HIWs) and their energy dissipation mechanisms over shoaling terrains (<300 m).

Additionally, the supplementary MODIS image shows that waves propagating near the seismic line 25 appear in the form of wave packets. We infer these small-amplitude high-frequency internal waves could be the result of the fission of these wave packets. The shoaling thermocline and gentle slope with a low internal Iribarren number support the point.

**RC5-3** I also think corroborating evidence such as satellite images or a tidal model indicating whether ISWs are anticipated to have been generated several days before at the Luzon Strait (i.e., phase of the spring-neap tidal cycle) is necessary. I am not convinced that the observations are not background, ubiquitous internal waves. Were the other survey lines similarly analyzed? If the other lines are analyzed, are these signals common (i.e., are they background noise)? I recommend using the other lines to form a background noise baseline and show whether the signals observed here actually deviate significantly from the background.

**AC** Thank you for your valuable comments. Firstly, we have supplemented satellite images and tidal models (lines 215-241, Figure R2 in this response letter). The results show that during the data acquisition period, ISW packets passed through line 25 (the studied line) and propagated northwest over the shoaling continental shelf. The tidal model indicates that this wave packet likely originated from neap tides in the Luzon Strait.

Secondly, to differentiate HIWs from background internal waves in seawater, we updated the seismic images (Figure 5, Figure R5 in this response letter). As shown in Figure 5a (Figure R5a in this response letter), background waves predominantly develop in the horizontal range of 70-80 km at depths above 400 m. In contrast, HIWs are mainly distributed on the left side of the seismic image (<20 km). Unlike background waves, the HIWs developing on the left side of the image exhibit high-frequency oscillations. In contrast, the background waves exhibit random oscillations with frequencies significantly lower than those of the HIWs observed on the left side of the seismic images (< 20 km). We also display background waves on other lines, as shown in Figure R22.

[Figure]

Figure R22 Background waves at the seismic line 28. The seabed is marked by the yellow solid line.

**RC5-4** Please add more discussion of the data collection and survey lines. For example, when is this specific line from? Where is it at in the spring-neap tidal cycle at the Luzon Strait where these waves presumably have been generated? Why was this the only line used? Was it the only line analyzed, or were other lines analyzed but there were no ISW signals?

**AC** Thank you for your valuable comments.

1. We have added more detailed information of the data collection and survey lines in the revised version (line 75-80). The Line 25 was collected on July 31.

2. We have supplemented satellite images (July 29- July 30) and tidal models (lines 215-241, Figure R2 in this response letter). The results show that during the data acquisition period, ISW packets passed through line 25 (the studied line) and propagated northwest over the shoaling continental shelf. The tidal model indicates that this wave packet likely originated from neap tides in the Luzon Strait.

3. In fact, as mentioned in our response to **RC3-1**, we observed large-amplitude ISWs on other seismic lines during the same cruise (Figures R14, R15 in this response letter). These symmetrical, large-amplitude ISWs are predominantly found in deeper waters at depths ranging from 350 to 700 m, with maximum amplitudes between 35 and 128 m, distant from the seabed topography. In contrast, our manuscript focuses on the high-frequency internal waves (HIWs) and their energy dissipation mechanisms over shoaling terrains (<300 m).

4. The reason for studying HIWs as follows: Bai et al. (2013, 2019) discovered that ISWs on the South China Sea shelf also undergo fission to produce high-frequency internal waves (HIWs) during shoaling and proposed a new hypothesis for the energy dissipation. Rippeth and Green (2020) pointed out that this finding is significant for understanding the dissipation mechanisms of ISWs. Therefore, whether HIWs, another product of fission, significantly enhance energy dissipation and mixing remains a critical question for comprehending the formation and evolution of wave packets and their role in ocean mixing and energy cascades. However, the energy dissipation mechanisms of IWs (ISWs) are complex and we have an incomplete understanding of the shoaling processes and ISWs' dissipation mechanisms. In this paper, we use the seismic method to study the high-frequency internal waves (HIWs) and their energy dissipation mechanisms over shoaling terrains, which improves our knowledge of ISW energy dissipation and their roles in improved mixing in the northern SCS.

**RC5-5** The seismic data processing discussion is convoluted and should be understandable to a

broader audience; please simplify it.

**AC** Thank you for your valuable comments. To address all reviewers' comments and improve the readability of the paper, we have revised the data processing section and provided explanations for technical terms (line 75-103). We have cited the relevant references. A detailed description of the seismic data processing can be found in Ruddick et al. (2009) and Holbrook et al. (2013).

**RC5-6** I think Sections 2.2 - 2.3 should be moved to the discussion instead of Data and methods. A lot of time is spent discussing different types of waves, which really is not relevant until the discussion and detracts from the methodology. The figures in these sections also might not be necessary and are a distraction from the observations.

**AC** Thank you for your valuable comments. We have deleted the Sections 2.2 - 2.3 about the discussion of different types of waves.

**Reference**

Acabado, C., Cheng, Y.-H., Chang, M.-H., and Chen, C.-C.: Vertical nitrate flux induced by Kelvin–Helmholtz billows over a seamount in the Kuroshio, Front. Mar. Sci, 8, 680729, 10.3389/fmars.2021.680729, 2021.

Aghsaee, P., Boegman, L., and Lamb, K. G.: Breaking of shoaling internal solitary waves, J. Fluid Mech., 659, 289-317, https://doi.org/10.1017/s002211201000248x, 2010.

Bai Y., Song H., Guan Y., Yang S., Liu B., Chen J., and Geng M.: Nonlinear internal solitary waves in the northeast South China Sea near Dongsha Atoll using seismic oceanography, Chinese Science Bulletin, 60, 944-951, https://doi.org/CNKI:SUN:KXTB.0.2015-10-010, 2015.

Bai, X. L., Liu, Z. Y., Li, X. F., Chen, Z. Z., Hu, J. Y., Sun, Z. Y., and Zhu, J.: Observations of high-frequency internal waves in the Southern Taiwan Strait, J. Coast. Res., 29, 1413-1419, https://doi.org/10.2112/JCOASTRES-D-12-00141.1, 2013.

Bai, X., Liu, Z., Zheng, Q., Hu, J., Lamb, K. G., and Cai, S.: Fission of shoaling internal waves on the northeastern shelf of the South China Sea, J. Geophys. Res.-Oceans, 124, 4529-4545, https://doi.org/10.1029/2018jc014437, 2019.

Bai, Y., Song, H., Guan, Y., and Yang, S.: Estimating depth of polarity conversion of shoaling internal solitary waves in the northeastern South China Sea, Cont. Shelf Res., 143, 9-17, https://doi.org/10.1016/j.csr.2017.05.014, 2017.

Bourgault, D., Blokhina, M. D., Mirshak, R., and Kelley, D. E.: Evolution of a shoaling internal solitary wavetrain, Geophys. Res. Lett., 34, L03601, https://doi.org/10.1029/2006GL028462, 2007.

Brandt, A., and Shipley, K. R.: Laboratory experiments on mass transport by large amplitude mode-2 internal solitary waves, Phys. Fluids, 26, 046601, https://doi.org/10.1063/1.4869101, 2014.

Chen, J. L., Yu, X., Chang, M. H., Jan, S., Yang, Y., and Lien, R. C.: Shear Instability and turbulent mixing in the stratified shear flow behind a topographic ridge at high Reynolds number, Front. Mar. Sci, 9, 829579, 10.3389/fmars.2022.829579, 2022.

Dickinson, A., White, N. J., and Caulfield, C. P.: Spatial variation of diapycnal diffusivity estimated

from seismic imaging of internal wave field, Gulf of Mexico, Journal of Geophysical Research: Oceans, 122, 9827-9854, https://doi.org/10.1002/2017JC013352, 2017.

Fan W., Song H., Zhang, K., Xu H., Gong Y., and Sun S.: Seismic oceanography study of internal solitary waves in the northeastern South China Sea Basin, Chinese Journal of Geophysics (in Chinese), 63, 2644-2657, https://doi.org/10.6038/cjg2020N0358, 2020.

Fan, W., Song, H., Gong, Y., Sun, S., Zhang, K., Wu, D., Kuang, Y., and Yang, S.: The shoaling mode-2 internal solitary waves in the Pacific coast of Central America investigated by marine seismic survey data, Continental Shelf Research, 212, https://doi.org/10.1016/j.csr.2020.104318, 2021.

Fan, W., Song, H., Gong, Y., Yang, S., and Zhang, K.: Regional study of mode-2 internal solitary waves at the Pacific coast of Central America using marine seismic survey data, Nonlin. Processes Geophys., 29, 141-160, https://doi.org/10.5194/npg-29-141-2022, 2022.

Fer, I., Nandi, P., Holbrook, W. S., Schmitt, R. W., and Páramo, P.: Seismic imaging of a thermohaline staircase in the western tropical North Atlantic, Ocean Science, 6, 621-631, 10.5194/os-6-621-2010, 2010.

Fu, K. H., Wang, Y. H., Laurent, L. C. S., Simmons, H. L., and Wang, D. P.: Shoaling of large-amplitude nonlinear internal waves at Dongsha Atoll in the Northern South China Sea, Cont. Shelf Res., 37, 1-7, https://doi.org/10.1016/j.csr.2012.01.010, 2012.

Geng, M., Song, H., Guan, Y., and Bai, Y.: Analyzing amplitudes of internal solitary waves in the northern South China Sea by use of seismic oceanography data, Deep-Sea Res. Part I-Oceanogr. Res. Pap., 146, 1-10, 10.1016/j.dsr.2019.02.005, 2019.

Geng, M., Song, H., Liu, S., Zhang, Y., Meng, L., Yang, B., Wang, L., Gu, Y., Rong, J., and Zhang, B.: Characteristics and migration of subaqueous sand dunes influenced by internal solitary waves in the Dongsha Region, Northern South China Sea, Geomorphology, 461, 109325, https://doi.org/10.1016/j.geomorph.2024.109325, 2024.

Gong, Y., Song, H., Zhao, Z., Guan, Y., and Kuang, Y.: On the vertical structure of internal solitary waves in the Northeastern South China Sea, Deep-Sea Res. Part I-Oceanogr. Res. Pap., 173, 103550, https://doi.org/10.1016/j.dsr.2021.103550, 2021a.

Gunn, K. L., Dickinson, A., White, N. J., and Caulfield, C.-c. P.: Vertical mixing and heat fluxes conditioned by a seismically imaged oceanic front, Front. Mar. Sci., 8, 697179, https://doi.org/10.3389/fmars.2021.697179, 2021.

Holbrook, W. S., Fer, I., Schmitt, R. W., Lizarralde, D., Klymak, J. M., Helfrich, L. C., and Kubichek, R.: Estimating oceanic turbulence dissipation from seismic images, J. Atmos. Ocean. Technol., 30, 1767-1788, https://doi.org/10.1175/jtech-d-12-00140.1, 2013.

Holbrook, W. S.: Thermohaline fine structure in an oceanographic front from seismic reflection profiling, Science, 301, 821-824, https://doi.org/10.1126/science.1085116, 2003.

Huang X., Song H., Guan Y., Geng M., and Wang Y.: Study of seawater seismic facies based on computational fluid dynamics., Chinese Journal of Geophysics (in Chinese), 61, 2892-2904,

https://doi.org/10.6038/cjg2018L0382, 2018.

Klymak, J. M., and Moum, J. N.: Oceanic isopycnal slope spectra. Part II: Turbulence, J. Phys. Oceanogr., 37, 1232-1245, https://doi.org/10.1175/JPO3074.1, 2007.

Krahmann, G., Papenberg, C., Brandt, P., and Vogt, M.: Evaluation of seismic reflector slopes with a Yoyo-CTD, Geophysical Research Letters, 36, 10.1029/2009gl038964, 2009.

Kuang Y., Wang Y., Song H., Guan Y., Fan W., Gong Y., and Zhang K.: Study of internal solitary wave packets in the northeastern South China Sea based on seismic oceanography and remote sensing, Chinese Journal of Geophysics (in Chinese), 64, 597-611, https://doi.org/10.6038/cjg2021N0399, 2021.

Lamb, K. G.: Internal wave breaking and dissipation mechanisms on the continental slope/shelf, Annu. Rev. Fluid Mech., 46, 231-254, https://doi.org/10.1146/annurev-fluid-011212-140701, 2014.

Liu, A. K., Chang, Y. S., Hsu, M. K., and Liang, N. K.: Evolution of nonlinear internal waves in the East and South China Seas, J. Geophys. Res.-Oceans, 103, 7995-8008, https://doi.org/10.1029/97JC01918, 1998.

Nakamura, Y., Noguchi, T., Tsuji, T., Itoh, S., Niino, H., and Matsuoka, T.: Simultaneous seismic reflection and physical oceanographic observations of oceanic fine structure in the Kuroshio extension front, Geophysical Research Letters, 33, https://doi.org/10.1029/2006GL027437, 2006.

Nandi, P., Holbrook, W. S., Pearse, S., Páramo, P., and Schmitt, R. W.: Seismic reflection imaging of water mass boundaries in the Norwegian Sea, Geophysical Research Letters, 31, https://doi.org/10.1029/2004GL021325, 2004.

Orr, M. H., and Mignerey, P. C.: Nonlinear internal waves in the south china sea: observation of the conversion of depression internal waves to elevation internal waves, J. Geophys. Res.-Oceans, 108, 3064, https://doi.org/10.1029/2001JC001163, 2003.

Rippeth, T., and Green, M.: Tides, the moon and the kaleidoscope of ocean mixing, in: Oceanography and Marine Biology: An Annual Review, edited by Hawkins S. J., Allcock A. L., et al, Taylor & Francis, 319-349, https://doi.org/10.1201/9780429351495-6, 2020.

Ruddick, B., Song, H., Dong, C., and Pinheiro, L.: Water column seismic images as maps of temperature gradient, Oceanography, 22, 192-205, https://doi.org/10.5670/oceanog.2009.19, 2009.

Sallarès, V., Biescas, B., Buffett, G., Carbonell, R., Dañobeitia, J. J., and Pelegrí, J. L.: Relative contribution of temperature and salinity to ocean acoustic reflectivity, Geophys. Res. Lett., 36, L00D06, https://doi.org/10.1029/2009gl040187, 2009.

Scotti, A., Beardsley, R. C., Butman, B., and Pineda, J.: Shoaling of nonlinear internal waves in Massachusetts Bay, J. Geophys. Res.-Oceans, 113, C08031, https://doi.org/10.1029/2008jc004726, 2008.

Sheen, K. L., White, N. J., and Hobbs, R. W.: Estimating mixing rates from seismic images of

oceanic structure, Geophys. Res. Lett., 36, L00D04, https://doi.org/10.1029/2009GL040106, 2009.

Shroyer, E. L., Moum, J. N., and Nash, J. D.: Observations of polarity reversal in shoaling nonlinear internal waves, J. Phys. Oceanogr., 39, 691-701, https://doi.org/10.1175/2008JPO3953.1, 2009.

Sinnett, G., Ramp, S. R., Yang, Y. J., Chang, M.-H., Jan, S., and Davis, K. A.: Large-amplitude internal wave transformation into shallow water, J. Phys. Oceanogr., 52, 2539-2554, https://doi.org/10.1175/JPO-D-21-0273.1, 2022.

Song, H., Chen, J., Pinheiro, L. M., Ruddick, B., Fan, W., Gong, Y., and Zhang, K.: Progress and prospects of seismic oceanography, Deep-Sea Res. Part I-Oceanogr. Res. Pap., 177, 10.1016/j.dsr.2021.103631, 2021a.

Song, H., Gong, Y., Yang, S., and Guan, Y.: Observations of internal structure changes in shoaling internal solitary waves based on seismic oceanography method, Front. Mar. Sci, 8, 10.3389/fmars.2021.733959, 2021b.

Sun S., Zhang K., and Song H.: Geophysical characteristics of internal solitary waves near the Strait of Gibraltar in the Mediterranean Sea, Chinese Journal of Geophysics (in Chinese), 62, 2622-2632, https://doi.org/10.6038/cjg2019N0079, 2019.

Tang, Q., Hobbs, R., Wang, D., Sun, L., Zheng, C., Li, J., and Dong, C.: Marine seismic observation of internal solitary wave packets in the northeast South China Sea, J. Geophys. Res.-Oceans, 120, 8487-8503, 10.1002/2015jc011362, 2015.

Tang, Q., Hobbs, R., Zheng, C., Biescas, B., and Caiado, C.: Markov Chain Monte Carlo inversion of temperature and salinity structure of an internal solitary wave packet from marine seismic data, J. Geophys. Res.-Oceans, 121, 3692-3709, https://doi.org/10.1002/2016jc011810, 2016.

Tang, Q., Wang, C., Wang, D., and Pawlowicz, R.: Seismic, satellite, and site observations of internal solitary waves in the NE South China Sea, Sci Rep, 4, 5374, https://doi.org/10.1038/srep05374, 2014.

Tang, Q., Xu, M., Zheng, C., Xu, X., and Xu, J.: A locally generated high-mode nonlinear internal wave detected on the shelf of the Northern South China Sea from marine seismic observations, J. Geophys. Res.-Oceans, 123, 1142-1155, 10.1002/2017jc013347, 2018.

Terletska, K., Choi, B. H., Maderich, V., and Talipova, T.: Classification of internal waves shoaling over slope-shelf topography, Russian Journal of Earth Sciences, 20, ES4002, https://doi.org/10.2205/2020ES000730, 2020.

Tsuji, T., Noguchi, T., Niino, H., Matsuoka, T., Nakamura, Y., Tokuyama, H., Kuramoto, S. i., and Bangs, N.: Two-dimensional mapping of fine structures in the Kuroshio Current using seismic reflection data, Geophys. Res. Lett., 32, n/a-n/a, 10.1029/2005gl023095, 2005.

Vlasenko, V., and Hutter, K.: Numerical experiments on the breaking of solitary internal waves over a slope–shelf topography, J. Phys. Oceanogr., 32, 1779-1793, 10.1175/1520-0485(2002)032<1779:Neotbo>2.0.Co;2, 2002.

Yang, S., Song, H., Coakley, B., Zhang, K., and Fan, W.: A mesoscale eddy with submesoscale

spiral bands observed from seismic reflection sections in the Northwind Basin, Arctic Ocean, J. Geophys. Res.-Oceans, 127, e2021JC017984, https://doi.org/10.1029/2021jc017984, 2022.

Yang, S., Song, H., Coakley, B., and Zhang, K.: Enhanced mixing at the edges of mesoscale eddies observed from high-resolution seismic data in the western Arctic Ocean, Journal of Geophysical Research: Oceans, 128, 10.1029/2023jc019964, 2023.

Zhang, K., Song, H., Coakley, B., Yang, S., and Fan, W.: Investigating Eddies From Coincident Seismic and Hydrographic Measurements in the Chukchi Borderlands, the Western Arctic Ocean, Journal of Geophysical Research: Oceans, 127, 10.1029/2022jc018453, 2022.

Zhao, Z., Klemas, V. V., Zheng, Q., and Yan, X.-H.: Satellite observation of internal solitary waves converting polarity, Geophys. Res. Lett., 30, https://doi.org/10.1029/2003GL018286, 2003.